# KPI-Chain: Multi-Agent Planning with Entity-Based Task Chaining for Reliable Recovery

## Abstract

Planning-based LLM agent frameworks promise flexible problem-solving through structured task decomposition, but they remain brittle: plans often fail silently, and existing approaches lack mechanisms for reliable recovery. We propose KPI-Chain, a multi-agent planning framework with a novel plan structure that embeds per-task key performance indicators (KPIs) based on typed entity extraction. This plan design—our core contribution—fundamentally improves agent reliability by making task specifications more precise and explicit upfront. In our formulation, each task explicitly defines expected entities (string, number, array, dict) to be extracted from its output. This structure drives multiple benefits throughout the system: it forces clearer, more specific task definitions during planning; it focuses extraction on only the relevant key information from tool responses rather than verbose outputs; it enables reasoning tasks to produce structured, targeted results; it supports efficient and precise memory management through typed entity storage; and critically, it makes failure root causes immediately identifiable when expected entities cannot be extracted. When KPIs are not met, the system automatically triggers continuation-based replanning with explicit failure feedback. To operationalize this plan structure, we introduce complementary components: an entity extractor for validating KPIs, a JSON-path memory system for typed entity storage and retrieval, MCP integration for standardized tool access, and chain-of-thought prompting for reasoning tasks. Across 5 challenging benchmarks, our KPI-Chain framework achieves higher success rates compared to existing agent architectures including ReAct and Plan-and-Execute. These results demonstrate that embedding entity-based KPIs directly into plan structure provides a foundation for building more reliable and adaptive LLM agent systems.

## 1 Introduction

Planning-based LLM agent frameworks have demonstrated remarkable capabilities in complex reasoning and task execution through structured task decomposition (Yao et al., 2023; Schick et al., 2024). However, their deployment in real-world multi-step scenarios reveals critical reliability issues: plans often fail silently without clear error signals, recovery mechanisms are ad-hoc or nonexistent, and task chaining frequently breaks due to inconsistent output formats (Qian et al., 2024; Wu et al., 2023). These failures stem not merely from execution problems, but fundamentally from how plans themselves are structured and specified.

Current agent frameworks like ReAct (Yao et al., 2023) and Plan-and-Execute approaches (Wang et al., 2023a) primarily focus on task orchestration but provide limited mechanisms for systematic failure detection and recovery. When intermediate tasks fail, these systems often continue execution with corrupted state, leading to cascading failures that are difficult to diagnose and correct. This brittleness significantly limits their applicability in production environments where reliability is paramount.

The challenge stems from several fundamental issues with existing plan representations. First, tasks are typically defined with natural language descriptions that lack explicit success criteria, making it difficult to determine programmatically whether a task has truly succeeded. Second, without structured specifications of what each task should produce, systems cannot validate outputs or manage state systematically, leading to fragile parameter passing between tasks. Third, when failures do occur, the absence of explicit failure indicators forces systems to resort to complete replanning rather than targeted recovery, resulting in inefficient use of previously successful work.

### 1.1 Our Approach and Contributions

We address these challenges by introducing a novel plan structure that embeds per-task key performance indicators (KPIs) based on typed entity extraction directly into task specifications. Our key insight is that explicitly defining expected entities (string, number, dict, array) for each task makes specifications more precise upfront and enables systematic validation throughout execution. A task succeeds only when all required entities are successfully extracted.

This plan structure drives multiple benefits: **(1) Precise task specifications** by forcing explicit output entity definitions upfront, **(2) Focused information extraction** by constraining extraction to relevant key information from tool responses, **(3) Structured reasoning outputs** enabling clean parameter passing, **(4) Efficient memory management** through typed entities with JSON-path references, and **(5) Immediate failure diagnosis** providing explicit, actionable feedback about what failed and why.

To operationalize this plan structure, we introduce complementary components: an Entity Extractor Agent that validates outputs against expected entities, a JSON-Path Global Memory System for typed entity storage and retrieval, a Re-planner Agent that generates continuation plans using explicit failure feedback, Model Context Protocol (MCP) integration for standardized tool access, and a Reasoning Agent using chain-of-thought prompting to produce structured entity outputs.

Empirical evaluation across multiple challenging benchmarks demonstrates substantially higher success rates compared to existing frameworks, establishing entity-based KPIs embedded in plan structure as a promising direction for reliable agent systems.

## 2 Related Work

Recent advances in LLM agent frameworks have focused on coordination mechanisms, action representations, and recovery strategies, but have overlooked how plan structure itself drives reliability improvements.

**Plan Representation and Task Orchestration.** ReAct (Yao et al., 2023) enables iterative reasoning and acting through interleaved thought-action loops but lacks structured task specifications with explicit success criteria. Plan-and-Execute approaches (Wang et al., 2023a) separate planning from execution but rely on natural language descriptions without formal output specifications, leading to fragile parameter passing and ambiguous success determination. Wang et al. (2024b) identified 14 failure modes in multi-agent systems, revealing that frameworks struggle with coordination precisely because they lack structured mechanisms for validating intermediate outputs. Unlike these approaches that treat task specification as an afterthought, our framework makes plan structure the central mechanism for reliability by embedding typed entity-based KPIs directly into task definitions.

**Action Representation and Execution.** CodeAct (Wang et al., 2024a) proposes executable Python code as a unified action space, demonstrating improved flexibility and composition. While CodeAct addresses how actions are represented and executed, it does not tackle what each action should produce or how to systematically validate success. Our entity-based KPI structure is complementary: regardless of whether actions execute as code, JSON, or tool calls, our plan structure explicitly defines expected typed entities for success determination.

**Validation and Recovery.** Reliability issues including hallucination, inconsistent outputs, and poor error handling remain critical bottlenecks (Zhang et al., 2024). Traditional approaches include self-verification (Madaan et al., 2024), multi-path reasoning (Yao et al., 2024), and ensemble methods (Wang et al., 2023b). SagaLLM (Chang & Geng, 2025) introduces transactional guarantees with compensation mechanisms and independent validation, but operates on existing task specifications without changing how tasks define success criteria. Our approach is complementary: embedding entity-based KPIs into plan structure provides explicit, actionable failure signals that could enhance transaction-based systems by making validation targets concrete and machine-verifiable from the outset. Unlike approaches that treat error detection and recovery as separate concerns, our entity-based KPIs integrate success measurement directly into task specification.

**Long-Context Reasoning.** The LOFT benchmark (Lee et al., 2024) evaluates models on needle-in-haystack problems requiring extraction from extremely long contexts, revealing that systems struggle to maintain context across reasoning steps. Our entity-based plan structure directly addresses this: by explicitly specifying which entities to extract at each step, our approach filters verbose responses to only relevant typed information, reducing context overhead while maintaining precise state tracking through JSON-path memory. This transforms long-context problems into structured state management, enabling efficient handling without overwhelming context windows.

Unlike existing frameworks focusing on coordination, action representations, or recovery mechanisms, our work establishes that embedding entity-based KPIs into plan structure provides a foundational mechanism for reliability: precise task specifications emerge naturally, validation becomes systematic rather than heuristic, and failure recovery leverages explicit knowledge of what succeeded and failed.

## 3 METHOD

Our KPI-Chain framework centers on a novel plan structure that embeds per-task entity-based KPIs directly into task specifications. This structure fundamentally changes how planning systems define, execute, and validate tasks. We describe the overall framework architecture (Section 3.1), then detail the plan structure design (Section 3.2), followed by how entity-based KPIs enable systematic validation (Section 3.3), structured state management through memory (Section 3.4), and reliable recovery through continuation-based replanning (Section 3.5).

### 3.1 FRAMEWORK OVERVIEW

Figure 1 illustrates the complete KPI-Chain architecture, which consists of three main layers designed to support and leverage the entity-based plan structure: Planning Layer: The Planner Agent generates structured execution plans where each task explicitly defines expected output entities with their types (string, number, array, dict). This upfront specification forces precise task definitions and enables systematic validation throughout execution. Execution Layer: Tasks are executed according to their type—tool calls invoke MCP servers while reasoning tasks use chain-of-thought prompting. Critically, the Entity Extractor Agent validates all tool call outputs against the expected entities defined in the plan structure, determining KPI success based on a calibrated confidence threshold (0.7). Reasoning tasks directly produce structured entity outputs without separate extraction. Reflection Layer: When KPIs are not met (entities missing or extracted with insufficient confidence), the Re-planner Agent analyzes the failure context and generates continuation plans that resume execution from the failure point while reusing successfully extracted entities from memory. Throughout execution, the JSON-Path Global Memory maintains both extracted entities and complete execution state (task statuses, plan structure), enabling precise parameter binding and intelligent recovery. The key innovation is how all components are designed around the entity-based plan structure: extraction targets come from expected entities, memory stores these typed entities, and replanning leverages explicit entity-level failure feedback.

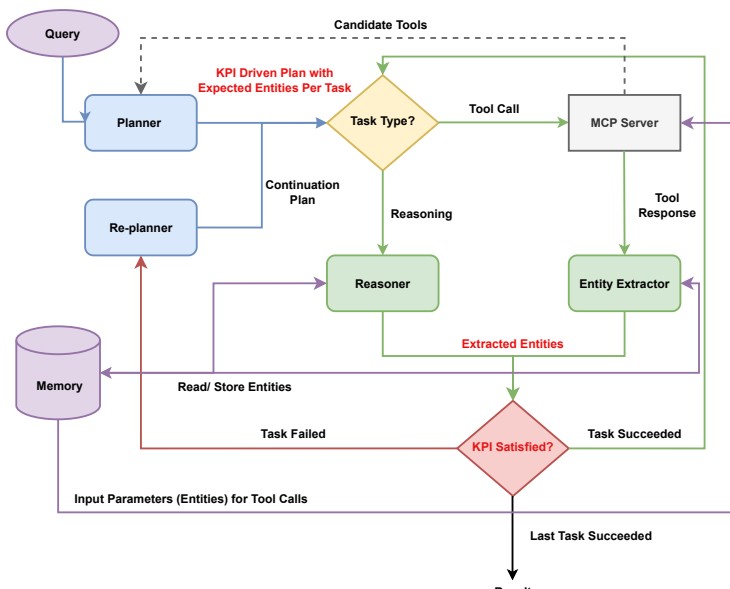

Figure 1: KPI-Chain framework overview showing the three-layer architecture: Planning, Execution with entity-based validation, and Reflection for failure recovery.

## 3.2 Plan Format Design

Listing 1: Plan Format

```
tasks:
  task_id: "Unique identifier"
  task_description: "Clear task description"
  task_type: "Tool call | Reasoning"
  tool_name: "Tool name"

  input_parameters:
    name: "param_name"
    type: "param_type"
    value: "Literal or <JSON_PATH>task.entity</JSON_PATH>"
    is_reference: boolean

  expected_output_entities:
    name: "entity_ID"
    type: "number, string, array or dict"
    description: "Output entity description"

  dependencies: ["task_id1", "task_id2"]
  execution_status: "pending/done/failed"
  execution_result: {}
```

The core of our approach is the plan structure itself. Unlike traditional approaches that specify tasks through natural language descriptions alone, our plans explicitly define success criteria through typed entity specifications. Figure 2 demonstrates this structure in action with a complete execution example.

The Planner Agent generates plans using a structured format that embeds KPI definitions directly into each task:

This structure provides several critical advantages. First, expected output entities serves as the explicit KPI specification–tasks succeed only when all required entities are extracted with sufficient confidence. Second, input parameters can reference previously extracted entities via JSON-path notation, enabling precise, type-aware parameter binding. Third, the explicit typing (string, number, array, dict) ensures semantic consistency across task chains. Fourth, execution status tracking enables the system to identify exactly which tasks succeeded or failed, supporting intelligent replanning.

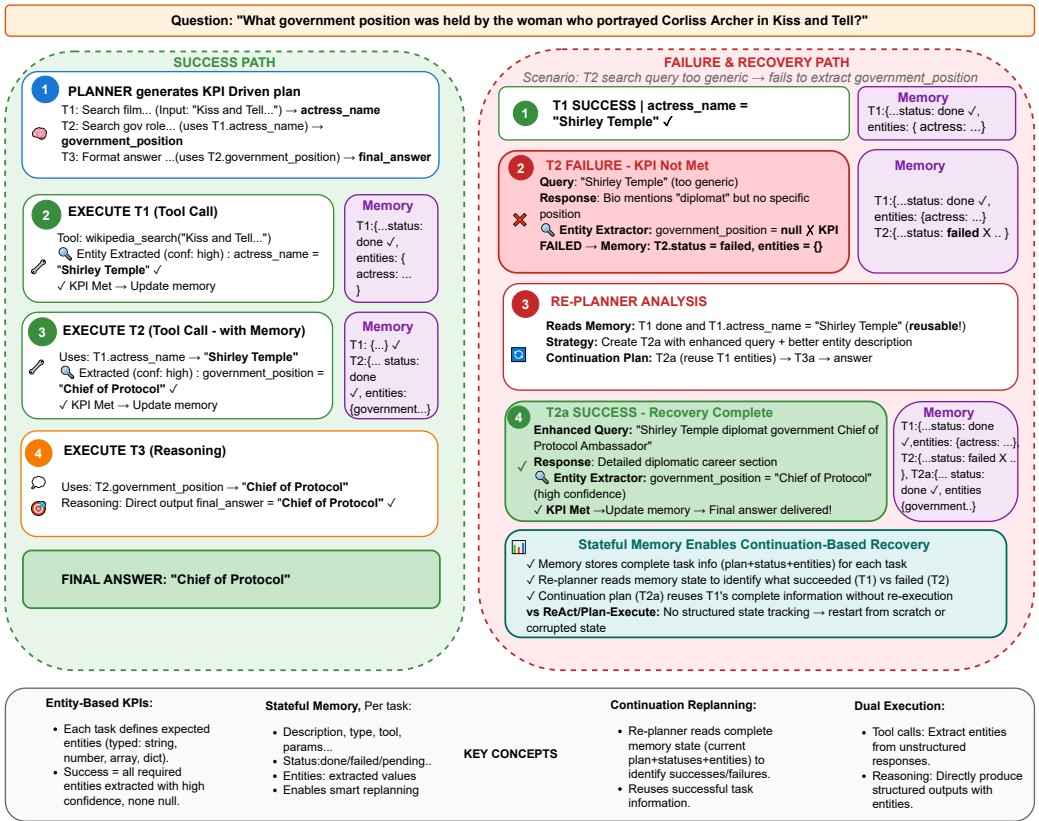

Figure 2: Execution example showing successful path (green) and failure recovery (red). When T2 fails to extract government position, the Re-planner generates continuation task T2a with an improved query for successful recovery.

As shown in Figure 2, this plan structure drives the entire execution flow: the planner creates tasks with expected entities, the executor validates against these entities, memory stores them with their types, and the re-planner uses execution status and entity information to generate targeted continuation plans.

## 3.3 Entity Extraction and Validation

The entity-based plan structure enables systematic validation through explicit success criteria. For tool call tasks, the Entity Extractor Agent validates the unstructured tool response against expected entities defined in the plan. For reasoning tasks, the Reasoning Agent directly produces both the reasoning process and structured entity outputs in a single LLM call, eliminating the need for separate extraction. For each expected entity, the extractor (or reasoning agent) determines a confidence score between 0 and 1 reflecting certainty in the extraction's correctness. The KPI success criterion is:

$$\text{KPI}_{\text{success}} = \forall e \in E : (\text{extracted}(e) \neq \text{None} \wedge \text{confidence}(e) \geq \theta)$$

where $\theta$ is a pre-tuned confidence threshold. This threshold is use-case dependent—different applications may require different precision-recall tradeoffs. In our experiments, we set $\theta = 0.7$ after evaluating performance at 0.1 intervals across benchmarks, but production systems should calibrate this threshold based on their specific reliability requirements. When any required entity is missing (None) or extracted with insufficient confidence, the system identifies this as KPI failure. The specific missing entities become explicit failure feedback for

targeted replanning—a direct benefit of the entity-based plan structure making expectations and outcomes equally explicit.

The framework uses a calibrated confidence threshold of 0.7 for entity validation. Entities that cannot be found are assigned a value of `None`, and if any required entity is `None`, the system automatically triggers replanning. For task outputs that exceed 3000 tokens, we split the content into manageable chunks and extract entities from each chunk separately. When the same entity is extracted multiple times across chunks, we resolve conflicts by selecting the candidate with the highest confidence score.

### 3.4 JSON-Path Global Memory System

Listing 2: Memory Architecture

```
1  {
2    "task_ID": {
3      "entity_ID": "typed value (string, number, array, dict)"
4    },
5    "current_plan": {
6      "tasks": [
7        {
8          "task_id": "T1",
9          "execution_status": "done",
10         "execution_result": {
11           "actress_name": "<ref:T1.actress_name>"
12         },
13         ...
14       }
15     ]
16   }
17 }
```

The global memory maintains structured state through two components: (1) extracted entities organized by task and entity name, and (2) the complete current plan with updated execution statuses and results. Memory Architecture:

Extracted entities enable efficient parameter binding via JSON-path references. The current plan with execution statuses provides comprehensive context for replanning: the re-planner sees which tasks succeeded or failed and which entities are available, enabling continuation plans that resume from failure points. As shown in Figure 2, this structure preserves complete execution history while supporting type-aware parameter flow.

### 3.5 Task Execution and Recovery

#### 3.5.1 Task Execution

Tool-based tasks integrate with Model Context Protocol (MCP) servers for standardized tool access. After execution, the Entity Extractor validates responses against expected entities, triggering replanning on KPI failure. Reasoning tasks use chain-of-thought prompting to directly produce structured outputs with expected entities in a single LLM call. Both task types apply the same KPI validation: all required entities must be present with correct types.

#### 3.5.2 Automatic Replanning and Recovery

When KPIs fail, the Re-planner Agent generates continuation plans resuming from failure points. As shown in Figure 2's recovery path (red), when T2 fails to extract government_position, the re-planner receives: (1) complete execution state showing T1 succeeded and T2 failed, (2) explicit failure feedback identifying the missing entity, and (3) original query context. The re-planner analyzes the root cause and generates continuation task T2a with enhanced query. Critically, T2a references T1's successfully extracted entity via <JSON_PATH>T1.actress_name</JSON_PATH>, preserving successful work while addressing the specific failure.

This continuation-based recovery leverages three benefits of entity-based plan structure: precise failure diagnosis (exactly which entities failed), selective reuse (successful entities preserved in memory), and targeted recovery (addressing specific missing entities rather than

restarting). Figure 2 shows successful recovery with T2a extracting government_position = "Chief of Protocol," enabling correct final answer generation.

# 4 EXPERIMENTAL SETUP

## 4.1 BENCHMARKS AND DATASETS

We evaluate our framework across five challenging benchmarks testing multi-step reasoning and long-context information retrieval. The LOFT (Long-Context Frontiers) benchmarks (Lee et al., 2024) emphasize needle-in-haystack problems where relevant information must be extracted from extensive contexts: **LOFT-MuSiQue** (100 questions, 2-4 hops requiring information synthesis), **LOFT-QAMPARI** (100 questions with multiple answers distributed across long documents), **LOFT-QUEST** (100 questions with underspecified reasoning requiring information acquisition), and **LOFT-TopiOCQA** (100 conversations with topic switching). We also evaluate on **HotpotQA** (Yang et al., 2018), a multi-hop reasoning benchmark requiring evidence synthesis (1,000 randomly sampled questions).

The LOFT benchmarks originally provide extensive context documents, but due to our model's 32K context limitation, we used a Wikipedia search tool that retrieved pages with very long contexts for these questions. This approach specifically exemplifies our entity extraction framework's core strength: the ability to address long-context challenges by extracting only relevant entities and information, significantly reducing context size for subsequent processing. These benchmarks test our entity-based plan structure's ability to extract relevant typed information from verbose tool responses, maintain structured state across reasoning steps, and recover from failures through explicit entity-level feedback.

## 4.2 BASELINE SYSTEMS

We compare against three primary baselines, all implemented using Qwen3-32B in thinking mode for fair comparison:

**ReAct** (Yao et al., 2023) synergizes reasoning and acting through interleaved thought-action loops, alternating between reasoning traces and actions but lacking systematic failure detection and structured state management.

**CodeAct** (Wang et al., 2024a) uses executable Python code as a unified action space, enabling flexible tool composition and dynamic action revision. While CodeAct improves action representation, it does not address structured task specification or systematic success validation—the core focus of our plan structure.

**Plan-and-Execute** (Wang et al., 2023a) separates planning from execution by generating multi-step plans upfront and executing them sequentially. This baseline represents our framework without entity-based KPIs: plans use natural language descriptions without explicit entity specifications, leading to unstructured state management and verbose failure feedback.

## 4.3 MODEL CONFIGURATION

All experiments use Qwen3-32B (Yang et al., 2025), a 32.8B parameter model with hybrid thinking/non-thinking modes and native 32K context window, exclusively in thinking mode to leverage enhanced reasoning capabilities. This model selection ensures fair comparison across all frameworks while demonstrating that our entity-based plan structure provides reliability improvements independent of model scale.

## 4.4 EVALUATION PROTOCOL

We employ task-specific evaluation metrics tailored to each benchmark type. For multi-hop QA (HotpotQA, MuSiQue), we use an LLM-as-judge approach with semantic comparison achieving >99% accuracy on validation samples. For multi-answer benchmarks (QAMPARI, QUEST), we measure recall of expected answers using semantic comparison. For multi-turn

conversations (TopiOCQA), we compute aggregated accuracy across conversation turns. All evaluation uses Qwen3-32B in thinking mode to maintain consistency. Additional metrics include tokens and LLM calls efficiency.

## 5 Results

### 5.1 Main Results

Table 1: Performance Comparison by Benchmark (Success Rate %)

| Benchmark | Sample Size | ReAct | CodeAct | Plan-Execute | KPI-Chain |
|---|---|---|---|---|---|
| LOFT-TopiOCQA | 100 | 30% | 33% | 31% | **41%** |
| LOFT-QAMPARI | 100 | 27% | 25% | 16% | **31%** |
| LOFT-QUEST | 100 | 15% | 18% | 13.5% | **25%** |
| LOFT-MuSiQue | 100 | 22% | 25% | 13% | **31%** |
| HotpotQA | 1,000 | 51% | 52% | 48% | **60%** |
| **Average** | | **29.0%** | **30.6%** | **24.3%** | **37.6%** |

Table 1 presents performance across all benchmarks. Our KPI-Chain framework achieves an average success rate of **37.6%** compared to ReAct at 29.0%, CodeAct at 30.6%, and Plan-and-Execute at 24.3%, representing a **23% relative improvement** over the strongest baseline (CodeAct).

The improvements are particularly pronounced in scenarios requiring complex entity tracking (LOFT-QUEST: +39% over CodeAct, +85% over Plan-Execute) and multi-hop reasoning with state management (HotpotQA: +15% over CodeAct, +25% over Plan-Execute). Notably, CodeAct's improved action representation provides modest gains over ReAct, but without structured entity-based KPIs, both approaches struggle with systematic validation and state management. Plan-and-Execute's poor performance demonstrates the critical importance of entity-based plan structure: without explicit entity specifications, even well-structured plans cannot systematically validate success or provide actionable failure feedback.

### 5.2 Failure and Recovery Analysis

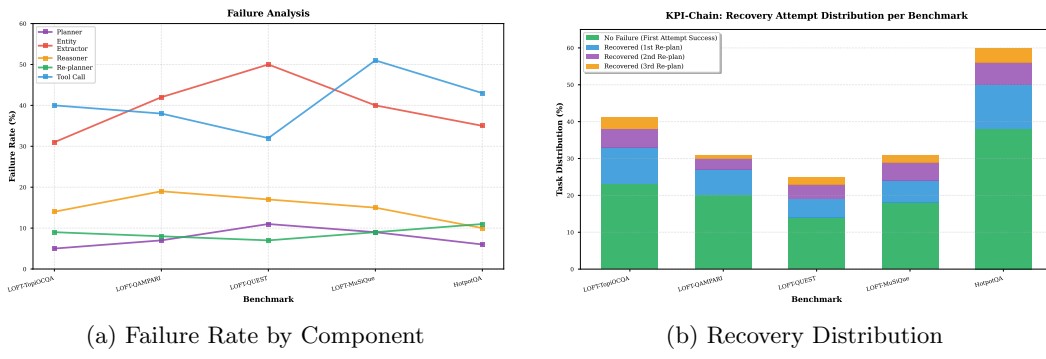

(a) Failure Rate by Component        (b) Recovery Distribution

Figure 3: Computational Analysis and Recovery Patterns

Figure 3a analyzes failure sources across components, revealing that entity extraction failures dominate LOFT benchmarks (31-50% failure rate), reflecting the challenge of extracting specific information from long, unstructured contexts—precisely the problem our entity-based plan structure addresses. Tool call failures are prominent in HotpotQA and LOFT-MuSiQue (43-51%), often due to query formulation issues that entity-based replanning addresses through failure feedback. Critically, planner and re-planner failures remain low (5-11%),

indicating that our entity-based plan structure enables robust plan generation and adaptation, while execution challenges arise primarily from extraction difficulty.

Figure 3b shows the distribution of recovery attempts across benchmarks, demonstrating how entity-based KPIs enable effective failure recovery. On HotpotQA, 38% of tasks succeed on first attempt while 50% succeed after 1-3 replanning attempts, validating that explicit entity-level failure feedback enables targeted recovery. The LOFT benchmarks show lower first-attempt success (14-23%) but substantial recovery through replanning (7-18% total recovered), confirming that entity-based KPIs provide actionable failure signals even in challenging long-context scenarios. Without entity-based KPIs (as in Plan-and-Execute), systems lack the structured feedback necessary for effective replanning, explaining Plan-and-Execute's consistently lower performance.

## 5.3 COMPUTATIONAL ANALYSIS

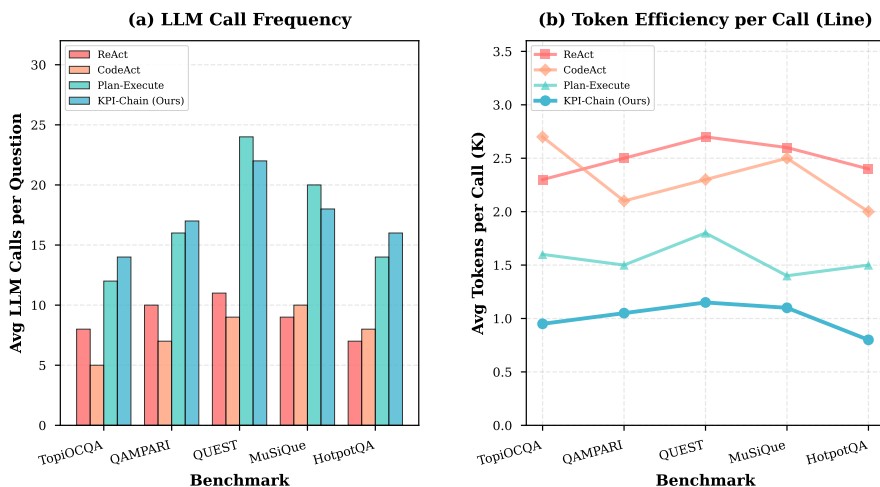

Figure 4: Computational efficiency analysis. (a) KPI-Chain exhibits higher LLM call frequency due to entity extraction and replanning. (b) KPI-Chain achieves superior token efficiency per call by extracting only typed entities rather than passing verbose responses.

Figure 4 presents computational efficiency metrics. Our approach exhibits higher LLM call frequency (Figure 4a) due to entity extraction and replanning overhead, representing the main computational tradeoff. However, Figure 4b reveals that KPI-Chain achieves superior **token efficiency per call**. This efficiency stems directly from our entity-based plan structure: by extracting only typed entities rather than passing verbose tool responses, we significantly reduce context overhead in subsequent calls.

## 5.4 ABLATION STUDIES

We conduct ablation studies to isolate the contribution of our entity-based plan structure and supporting components.

**Impact of Entity-Based Plan Structure.** Removing expected output entities eliminates the foundation for entity extraction, structured memory, and targeted replanning—effectively reducing our system to Plan-and-Execute. Table 1 shows this achieves only 24.3% success rate versus 37.6% for KPI-Chain, a 35% relative degradation that directly validates our core contribution.

**Impact of Continuation-Based Replanning.** Figure 3b shows 14-62% of successful executions required replanning. Without the re-planner, these tasks would fail. On HotpotQA, removing replanning reduces success from 60% to approximately 38%, a 37% degradation, validating that entity-based failure feedback enables targeted recovery rather than restarts from scratch.

**Impact of Structured Memory.** Figure 4b shows our approach uses 39-69% fewer tokens per call than ReAct, directly attributable to passing compact typed entities rather than verbose tool responses. This efficiency compounds across multi-hop chains, enabling coherent state maintenance where baselines exceed context limits.

**Component Synergy.** Figure 3a shows low planner/re-planner failure rates (5-11%) versus substantial entity extraction failures (31-51%), indicating our plan structure produces robust plans while execution challenges arise from extracting information from unstructured responses. This validates making entity specifications explicit upfront for systematic validation and targeted recovery.

## 6 DISCUSSION

Our experimental results demonstrate that embedding entity-based KPIs directly into plan structure provides a robust foundation for reliable LLM agent systems. The 23% relative improvement over the strongest baseline and 35% degradation when removing entity specifications validate that explicit, typed entity definitions are fundamental to systematic task validation and state management. The framework excels in scenarios requiring state tracking across reasoning steps, reliable parameter passing, recovery from intermediate failures, and handling needle-in-haystack problems in long contexts.

Analysis of failure patterns reveals important characteristics. Entity extraction failures dominate LOFT benchmarks (31–50%), reflecting the challenge of extracting specific information from unstructured tool responses. However, low planner and re-planner failure rates (5–11%) indicate the entity-based plan structure itself is robust—execution challenges arise primarily from extraction difficulty. The substantial recovery through replanning (14–62% of successful tasks) validates that entity-based failure feedback enables targeted continuation plans rather than complete restarts, as the system knows precisely which entities failed to extract.

The primary limitation is computational overhead: entity-based plans require additional LLM calls for extraction and replanning, resulting in higher call frequency despite superior token efficiency per call. This tradeoff may be acceptable for reliability-critical applications but could be prohibitive in resource-constrained scenarios. The approach is also less suitable for highly creative tasks where rigid entity constraints might limit desirable variability, and our current design assumes upfront planning is possible, which may not hold when task structure depends on information discovered during execution.

Several promising directions could extend the framework's capabilities. First, supporting exploratory planning where plan structure depends on unknown information would enable applications requiring iterative discovery before structured planning. Second, fine-tuning specialized small models for each agent component could substantially reduce computational costs while maintaining performance. Third, extending to multi-modal entity extraction would enable handling diverse input types (images, audio, video) while maintaining the same entity-based validation paradigm.

## 7 CONCLUSION

We introduced KPI-Chain, a multi-agent planning framework centered on a novel plan structure that embeds per-task entity-based KPIs directly into task specifications. By explicitly specifying expected typed entities, our approach forces precise task definitions, focuses extraction on relevant information, enables structured state management, and provides actionable failure feedback for targeted recovery. Evaluation across five challenging benchmarks demonstrates consistent improvements over existing frameworks, with particularly strong gains in multi-hop reasoning and long-context scenarios. This work establishes that embedding structured success criteria into plan representation provides a principled foundation for building more reliable and adaptive LLM agent systems.

## REPRODUCIBILITY STATEMENT

To ensure reproducibility of our results, we provide comprehensive implementation details throughout this submission. The main paper (Section 4) specifies the model architecture (Qwen3-32B in thinking mode), all benchmarks used (LOFT-TopiOCQA, LOFT-QAMPARI, LOFT-QUEST, LOFT-MuSiQue, and HotpotQA), sample sizes, and evaluation metrics. Section 3.3 details the confidence threshold and the systematic methodology used to determine it. The entity types, KPI validation formulas, and memory structures are formally defined in Sections 3.3 and 3.4.

The appendix contains all prompts used for each agent component (Planner, Entity Extractor, Reasoner, and Re-planner), complete with system instructions and few-shot examples. Baseline implementations (ReAct, CodeAct, Plan-and-Execute) are described with sufficient detail to enable fair reproduction, including how we adapted CodeAct to our evaluation setting.

Section 4.4 describes our evaluation protocol in detail, including the LLM-as-judge approach for multi-hop QA with validation accuracy metrics, and semantic comparison methods for multi-answer benchmarks.

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

## A  Declaration of Large Language Model Use

In accordance with ICLR 2026 submission guidelines, we declare the use of large language models during the preparation of this manuscript. Specifically, we used Claude (Anthropic) for the following purposes:

**Writing Aid and Polish:** Claude was used to improve the clarity, flow, and academic writing style of the manuscript. This included assistance with sentence structure, paragraph organization, and ensuring consistent terminology throughout the paper.

**Retrieval and Discovery:** Claude was employed to help identify and discover related work in multi-agent systems, entity extraction, and long-context reasoning. This assisted in ensuring comprehensive coverage of relevant literature and proper positioning of our contribution within the existing research landscape.

All technical contributions, experimental design, implementation, evaluation, and core insights presented in this work are the original contributions of the authors. The use of Claude was limited to writing assistance and literature discovery, and did not involve generation of technical content, experimental results, or novel ideas.

## B  Agent Prompt Templates

This section provides placeholders for the core prompt templates used for each specialized agent in the KPI-Chain framework. Each agent uses a two-part prompt structure: a system prompt that defines the agent's role and capabilities, and a user prompt that provides the specific task context.

### B.1  Planner Agent Prompt

#### B.1.1  System Prompt

```
You are a planning assistant that breaks down queries into structured execution plans.

CRITICAL: Your role is to GENERATE A PLAN, not to solve the query. Do not attempt to answer questions or
    provide solutions - only create the execution plan that will be used to solve the query.

# TASK TYPES

## Tool Call
Executes a tool and automatically extracts specific values in ONE task:
- Tool runs with input_parameters -> returns raw response
```

```
648
649   - Entity extraction processes raw response -> extracts values from expected_output_parameters
      - Only extracted values are stored (raw response discarded)
650
651   Extract directly: `expected_output_parameters: [{{name: person_birth_date, description: "Birth date
          extracted from Wikipedia"}}]`
652   Never store raw then extract separately: Don\'t create T1 with `raw_content` output, then T2 to extract
          from it
653
654   ## Reasoning
655   For LLM-based tasks such as: analysis, comparisons, calculations, decisions, summarization, text
          generation, classification, formatting, or any other task an LLM can perform. NOT for extracting
656       fields from tool responses (that\'s done automatically in Tool Call tasks).
657
      # INSTRUCTIONS
658
659   1. **Break down the query** into Tool call (gather data) and Reasoning (analyze/process with LLM) tasks
660
      2. **For each task define:**
661   - task_id (T1, T2, T3...)
      - task_description (use parameter names, not JSON paths)
662   - task_type ("Tool call" or "Reasoning")
      - tool_name (tool name or "")
663   - input_parameters (empty list [] if none)
664   - expected_output_parameters
      - dependencies (task_ids or empty list [])
665
666   3. **Tool call input_parameters:**
      - CRITICAL: Must EXACTLY match the tool\'s signature - use the exact parameter names, types, and structure
667       defined in the tool
668   - Include only parameters that exist in the tool definition
      - Include all required parameters
669   - Case-sensitive names
670   - Do NOT add, remove, or rename any parameters from the tool\'s signature
671
      4. **Tool call expected_output_parameters:**
672   - Define what to EXTRACT (not raw data)
      - Use descriptive names: `einstein_birth_year`, `is_raining`, `user_email`
673   - Never: `raw_data`, `response`, `content`
674   - Write clear descriptions to guide extraction
675
      5. **Referencing previous task outputs:**
676   - Tasks can ONLY access previous outputs via input_parameters
      - Use: `value: <JSON_PATH>task_id.param_name</JSON_PATH>` with `is_reference: true`
677   - Arrays: `<JSON_PATH>task_id.param_name[*]</JSON_PATH>`
      - No string interpolation: [INCORRECT] `"search <JSON_PATH>T1.name</JSON_PATH>"`, [CORRECT] `<JSON_PATH>T1
678       .name</JSON_PATH>`
679   - Empty input_parameters = task cannot access ANY previous outputs
680   - Never use internal knowledge - only data from input_parameters
681
      6. **Final task:**
682   - Must output single parameter: `final_answer`
      - Must have input_parameters if needs previous task data
683   - Answer should be clear and concise
684
      # OUTPUT FORMAT
685   ```yaml
686   tasks:
      - task_id: string
687     task_description: |-
          # Clear description
688     task_type: Tool call or Reasoning
        tool_name: string
689     input_parameters:
690       - name: param_name
            type: param_type
691         value: |-
              # Literal or <JSON_PATH>task_id.param</JSON_PATH>
692         is_reference: true | false
693     expected_output_parameters:
          - name: param_name
694         description: |-
              # What to extract/produce
695         type: param_type
        dependencies: []
696     execution_result: {{}}
697   ```
698
699
700   # KEY RULES
      - You are ONLY creating a plan, not solving the query
701   - No separate extraction tasks - extract in Tool call itself
      - Tool input parameters must match tool signature EXACTLY (same names, types, structure)
```

```
- Extract specific values, never store raw responses
- Tasks need input_parameters to access previous outputs
- No internal knowledge - only use input_parameters data
- No string interpolation in values
- Final task needs input_parameters if using previous data
- Output only YAML plan, no explanations
```

### B.1.2 USER PROMPT

```
## Available tools:
```yaml
{tools}
```

## Query:
{query}

## Output:
```

## B.2 RE-PLANNER AGENT PROMPT

### B.2.1 SYSTEM PROMPT

```
You are a re-planning assistant that adapts plans when tasks fail.

CRITICAL: Your role is to GENERATE A CONTINUATION PLAN, not to solve the query. Do not attempt to answer
    questions or provide solutions - only create the continuation plan that will be used to recover from
    the failure and solve the query.

# YOUR ROLE
Create continuation plan from failure point while preserving successful work. Get back on track to answer
    the original query.

# TASK TYPES

## Tool Call
Executes tool AND extracts values in ONE task. If extraction failed, improve descriptions in the Tool call
     itself - don't add separate extraction task.

## Reasoning
For LLM-based tasks such as: analysis, comparisons, calculations, decisions, summarization, text
    generation, classification, formatting, or any other task an LLM can perform. NOT for extracting
    fields from tool responses (that's done automatically in Tool Call tasks).

# INSTRUCTIONS

1. **Diagnose failure** from feedback:
   - Why failed? (wrong tool, extraction failed, missing input, tool error)
   - What outputs missing?
   - Which tasks depend on them?
   - What alternatives work?

2. **Choose strategy:**
   - Extraction failed -> Improve output parameter descriptions
   - Wrong tool -> Switch to different tool
   - Missing input -> Add preceding tasks
   - Tool error -> Use fallback approach

3. **Create continuation tasks** from failure point:
   - Continue task sequence with alternate id (if T3 failed - T3a, T4a...)
   - Task id shouldn't be task already used in previous plan
   - Modifies only the failed task and any dependent future tasks
   - Uses alternative approaches to resolve or work around the failure
   - Maintains the same end goal as the original plan

4. **Tool call input_parameters:**
   - CRITICAL: Must EXACTLY match the tool's signature - use the exact parameter names, types, and
     structure defined in the tool
   - Include only parameters that exist in the tool definition
   - Include all required parameters
   - Case-sensitive names
   - Do NOT add, remove, or rename any parameters from the tool's signature

5. **Tool call expected_output_parameters:**
   - Define what to EXTRACT
   - Use descriptive names
```

```
    - If extraction failed before, make descriptions MORE specific and detailed

6. **Reference previous outputs:**
    - Use: `value: <JSON_PATH>task_id.param_name</JSON_PATH>` with `is_reference: true`
    - No string interpolation
    - Empty input_parameters = cannot access previous outputs
    - Only use input_parameters data, not internal knowledge
    - If task needs previous data, MUST be in input_parameters

7. **Final task:**
    - Must output single parameter: `final_answer`
    - Must have input_parameters if needs previous task data
    - Answer should be clear and concise

# OUTPUT FORMAT
```yaml
tasks:
  - task_id: string
    task_description: |-
      # Clear description
    task_type: Tool call or Reasoning
    tool_name: string
    input_parameters:
      - name: param_name
        type: param_type
        value: |-
          # Literal or <JSON_PATH>task_id.param</JSON_PATH>
        is_reference: true | false
    expected_output_parameters:
      - name: param_name
        description: |-
          # What to extract/produce
        type: param_type
    dependencies: []
    execution_result: {{}}
```

# KEY RULES
- You are ONLY creating a continuation plan, not solving the query
- Preserve completed tasks
- No separate extraction tasks - improve in Tool call
- Tool input parameters must match tool signature EXACTLY (same names, types, structure)
- Extract values, never raw responses
- Diagnose before fixing
- If extraction failed, make descriptions more specific
- Tasks need input_parameters to access previous outputs
- No internal knowledge - only input_parameters
- No string interpolation
- Final task needs input_parameters if using previous data
- Output only YAML plan
- Prefer atomic types (string, int, bool, float)
```

### B.2.2 USER PROMPT

```
## List of available tools:
```yaml
{tools}
```

## Input:

### Original Query:
{query}

### Current Plan (with failure):
```yaml
{current_plan}
```

### Failed Task ID:
{failed_task_id}

### Failure Feedback:
{feedback}

## Output:
```

## B.3 ENTITY EXTRACTOR AGENT PROMPT

### B.3.1 SYSTEM PROMPT

```
You are an expert system designed for high-accuracy entity extraction from API responses.

## Goal
Your purpose is to parse a given API RESPONSE generated by the TASK DESCRIPTION to find and extract
    specific entities.

## Instructions
1. Carefully review the TASK DESCRIPTION understand what is being asked and how this data was generated
2. Examine the API RESPONSE available to you to identify the most relevant information or information that
    can help with solving this item
3. Follow a step-by-step reasoning process based on the provided input to derive the requested outputs
4. Assign a confidence score between 0.0 and 1.0 for your extraction, where:
   - 0.0-0.3: Very low confidence (highly uncertain, likely incorrect)
   - 0.3-0.5: Low confidence (uncertain, may be incorrect)
   - 0.5-0.7: Medium confidence (somewhat confident, but not certain)
   - 0.7-0.9: High confidence (confident in extraction)
   - 0.9-1.0: Very high confidence (extremely confident, clearly stated)

## Input Format
TASK DESCRIPTION:
The definition of the API call that generated this response

API RESPONSE:
The API response for this specific task

ENTITIES TO EXTRACT:
A list of entities to extract from the API response, where each entity has a name and description

## Output Format
Your response **must** be in the following valid YAML format.
```yaml
confidence_score: <float between 0.0 and 1.0>
extracted_entities:
  entity_name: <extracted_value> or `null` # `null` in case the entity does not mentioned in the API
      response
entities_summary: |-
  Reasoning behind this decision and confidence score
```

## Important:
- CRITICAL: You must answer in the requested output format
- The confidence_score must be a numeric value between 0.0 and 1.0
- If no information found or something is wrong, return `null` for the expected entity no matter what is
    the output type, and assign an appropriately low confidence score
- Wrap strings with quotes and escape characters in case of need
- Make sure your response is fully-completed, meaning capture all required information for this entity
```

### B.3.2 USER PROMPT

```
API RESPONSE:
{api_response}

TASK DESCRIPTION:
{api_description}

ENTITIES TO EXTRACT:
```yaml
{entities}
```
```

## B.4 REASONING AGENT PROMPT

### B.4.1 SYSTEM PROMPT

```
You are a reasoning engine responsible for executing reasoning tasks within a task execution plan.
Your role is to process a specific reasoning task, apply logical thinking to the input parameters, and
    produce the expected output parameters.

## Instructions

1. Carefully review the task description and understand what is being asked
```

```
2. Examine all input parameters available to you
3. Follow a step-by-step reasoning process based on the provided input to derive the requested outputs
4. Ensure that all expected output parameters are generated
5. If you cannot produce an expected output parameter, set its value to `null`
6. Format your response as a justification string followed by a structured object
7. Always include all expected output parameters in your response, even if some have 'null' values

## Input Format

```yaml
task_description: "string"
input_parameters:
  - name: "param_name1"
    value: "value1 or <param_name_from_previous_task>"
    type: "string"
  - name: "param_name2"
    value: "value2 or <param_name_from_previous_task>"
    type: "string"
expected_output_parameters:
  - name: "output1"
    description: "Description of the first expected output"
    type: "string"
  - name: "output2",
    description: "Description of the second expected output",
    type: "string"
```

## Output Format
 Justifications:
 1. ...
 2. ...

 ...

 N. ...

# Disclaimers:
# For **long/complex string** items use the following literal block scalars format: ```
# key: |
#    <STRING_VALUE_WITHOUT_QOUTES>```
# For list of strings:
# ```
# list_key:
#    - |
#      <STRING_VALUE_WITHOUT_QOUTES>```
# For empty list use the following format `key: []`
# For null value use the following format `key: null`
# You MUST escape (\) special characters ({,:,",-,#,|,',',\)
#
# Schema:
execution_result:  # Results of the execution including status and outputs
  status: |
    $STRING_VALUE # completed | failed (in case at least one value can't be found)
  outputs:  # Key-Value output values from the execution, value should be null for the key that can't be
    resolved
    $OBJECT_VALUE
execution_details:  # Details about the execution process including reasoning
  reasoning_steps: []  # Sequential steps of reasoning that led to the execution result

## Note:
- CRITICAL: You must answer in the requested output format
- IMPORTANT: Do not take hidden assumption, rely on the provided input
- If no information found, you cannot determine the result or something is wrong, return `null` for the
    expected output for any data type and set status to `failed`
- Verify the generated text fits to original input
- Think step-by-step before generating the yaml answer
```

### B.4.2  USER PROMPT

```
## Input
```yaml
{reasoning_task} # the reasoning task and its values
```
```

### B.5 Prompt Usage and Execution Flow

This section describes how each agent prompt is utilized within the KPI-Chain framework, including input preparation, execution context, and output processing.

#### B.5.1 Planner Agent Usage

**Invocation Context:** The Planner Agent is invoked at the beginning of query processing to generate the initial execution plan.

**Input Preparation:**

- {`tools`}: Serialized YAML representation of all available MCP tools with their signatures, parameter types, and descriptions
- {`query`}: The original user question or task to be solved

**Execution:** The system prompt establishes the planner's role and constraints, while the user prompt provides the specific query and tool catalog. The model generates a structured YAML plan following the schema in Listing 1.

**Output Processing:**

- Parse YAML output into task objects
- Validate task structure (required fields, dependency consistency)
- Initialize execution status for all tasks as "pending"
- Store plan in execution context for task orchestration

**Error Handling:** If YAML parsing fails or the plan structure is invalid, the planner is re-invoked with error feedback up to 3 retry attempts.

#### B.5.2 Entity Extractor Agent Usage

**Invocation Context:** The Entity Extractor is invoked ONLY after tool call executions to extract expected entities from tool responses. Reasoning tasks produce entities directly and do not use the Entity Extractor.

**Input Preparation:**

- {`api_response`}: Raw output from MCP tool execution
- {`api_description`}: The task description from the plan explaining what the tool call was intended to accomplish
- {`entities`}: YAML list of expected output parameters with names, types, and descriptions

**Execution:** The extractor uses the task description to understand context and the entity descriptions to guide extraction from unstructured tool outputs.

**Output Processing:**

- Parse YAML output containing confidence score (0.0-1.0) and extracted entities
- Evaluate KPI: success if all entities $\neq$ null and confidence score $\geq 0.7$
- Store extracted entities in JSON-path global memory at `task_id.entity_name`
- If KPI fails, trigger re-planning with failure feedback including the confidence score

#### B.5.3 Re-planner Agent Usage

**Invocation Context:** The Re-planner is invoked when a task fails: either when a tool call's KPI validation fails (entities not extracted with sufficient confidence), or when a reasoning task returns status "failed".

**Input Preparation:**

- {`tools`}: Same tool catalog as provided to Planner
- {`query`}: Original user query
- {`current_plan`}: The full execution plan including completed tasks and the failed task
- {`failed_task_id`}: Identifier of the task that failed
- {`feedback`}: Detailed failure information including:
  - For tool calls: Which entities were not extracted, confidence scores, entity extractor's reasoning
  - For reasoning tasks: Reasoning agent's failure explanation
  - Tool error messages (if applicable)

**Execution:** The re-planner analyzes the failure context and generates continuation tasks that:

- Use alternative task IDs (e.g., T3a if T3 failed)
- For tool call failures: Incorporate more specific entity descriptions or switch to alternative tools
- For reasoning failures: Adjust input parameters, break down complex reasoning, or provide additional context
- Add prerequisite tasks if required data was missing
- Maintain references to successfully extracted entities in memory

**Output Processing:**

- Parse YAML continuation plan
- Merge continuation tasks into execution plan after failed task
- Remove or modify downstream tasks affected by the failure
- Resume execution from the first continuation task

**Retry Limit:** Re-planning is limited to 3 attempts per task to prevent infinite loops. If a task fails after 3 re-planning attempts, the entire execution fails.

### B.5.4  REASONING AGENT USAGE

**Invocation Context:** The Reasoning Agent is invoked for tasks with `task_type: "Reasoning"`, which require cognitive processing such as analysis, comparison, synthesis, or decision-making.

**Input Preparation:**

- {`reasoning_task`}: YAML object containing:
  - `task_description`: Description of the reasoning task
  - `input_parameters`: List of input values, resolved from literals or JSON-path references to previous task outputs
  - `expected_output_parameters`: List of outputs to generate with descriptions

**Parameter Resolution:** Before invoking the Reasoning Agent, the system resolves all JSON-path references in input parameters:

- `<JSON_PATH>T1.entity_name</JSON_PATH>` $\rightarrow$ lookup value in global memory at `T1.entity_name`
- `<JSON_PATH>T2.list_entity[*]</JSON_PATH>` $\rightarrow$ retrieve entire array from memory
- Replace reference with actual value in input parameters

**Execution:** The Reasoning Agent performs step-by-step logical reasoning using the provided inputs and generates all expected output parameters in structured YAML format. Unlike tool calls, the Reasoning Agent is responsible for both reasoning AND producing the entities directly in the expected format.

**Output Processing:**

- Parse YAML output containing execution result and reasoning steps
- Extract outputs from `execution_result.outputs`
- Check status field:
  - If status = "completed": Validate all expected entities are present and non-null, then store entities directly in global memory
  - If status = "failed": Trigger re-planning with reasoning failure feedback
- Store reasoning steps in execution trace for debugging

**Entity Production:** The Reasoning Agent's output format directly provides entities without additional extraction:

```
1  execution_result:
2    status: completed
3    outputs:
4      entity_1: "value_1"
5      entity_2: "value_2"
```

These outputs are stored directly in global memory at `task_id.entity_name`, enabling efficient parameter passing to subsequent tasks without intermediate extraction steps.

### B.5.5  TASK TYPE EXECUTION SUMMARY

**Tool Call Tasks:**

1. Execute MCP tool with resolved input parameters
2. Receive unstructured tool response
3. Invoke Entity Extractor to extract expected entities
4. Validate KPI (confidence $\geq 0.7$, all entities $\neq$ null)
5. Store extracted entities in global memory or trigger re-planning

**Reasoning Tasks:**

1. Resolve input parameter JSON-path references
2. Invoke Reasoning Agent with task description and inputs
3. Reasoning Agent produces structured YAML with entities
4. Validate status and entity presence (no separate extraction)
5. Store entities directly in global memory or trigger re-planning

This dual-path approach optimizes for each task type: tool calls require entity extraction from unstructured responses, while reasoning tasks leverage the LLM's ability to produce structured outputs directly.

### B.5.6  GLOBAL MEMORY AND JSON-PATH RESOLUTION

Throughout execution, the system maintains a hierarchical JSON structure in global memory with two components: (1) extracted entities organized by task and entity name, and (2) the complete current plan with execution statuses.

**Storage:** After successful entity extraction (tool calls) or direct entity production (reasoning tasks), the memory is updated as:

```
1  {
2    "task_id": {
3      "entity_name": value
4    },
5    "current_plan": {
6      "tasks": [
7        {
8          "task_id": "T1",
9          "task_description": "...",
10         "execution_status": "done",
11         "execution_result": {
12           "actress_name": "<ref:T1.actress_name>"
13         }
14       },
15       {
16         "task_id": "T2",
17         "task_description": "...",
18         "execution_status": "failed",
19         "execution_result": {}
20       }
21     ]
22   }
23 }
```

The `current_plan` component provides complete execution context for replanning: the re-planner sees which tasks succeeded or failed and which entities are available, enabling generation of continuation plans that resume from failure points while preserving successful work.

**Retrieval:** When executing a task with JSON-path references:

1. Parse input parameter value for `<JSON_PATH>` tags
2. Extract path: `task_id.entity_name` or `task_id.entity_name[*]`
3. Query global memory using the path (from the entities component)
4. Replace `<JSON_PATH>` with retrieved value
5. Pass resolved value to tool or reasoning agent

**Type Preservation:** The memory system preserves entity types:

- Strings remain strings
- Numbers remain numeric types
- Arrays remain arrays (with `[*]` notation)
- Dictionaries remain nested objects

This typed memory ensures that parameter binding maintains semantic correctness across task chains, while the plan state enables intelligent replanning with full execution history.

## C  COMPLETE EXECUTION TRACE EXAMPLE WITH RE-PLANNING

This section provides a detailed, step-by-step execution trace for a HotpotQA question demonstrating the full KPI-Chain framework in action, including planning, execution, entity extraction, failure detection, re-planning, and memory management throughout the entire process.

### C.1  QUERY

**Question:** "What government position was held by the woman who portrayed Corliss Archer in the film Kiss and Tell?"

**Expected Answer:** Chief of Protocol

**Question Type:** Multi-hop reasoning requiring (1) identifying the actress from the film, then (2) finding her government position.

## C.2 Step 1: Initial Plan Generation

**Planner Agent Invocation:**

*Input:*

```
1  tools:
2    - name: wikipedia_search
3      description: Search Wikipedia for information
4      parameters:
5        - name: query
6          type: string
7          required: true
8
9  query: "What government position was held by the woman who portrayed
10         Corliss Archer in the film Kiss and Tell?"
```

*Output (Generated Plan):*

```
1  tasks:
2    - task_id: T1
3      task_description: |-
4        Search for information about the film "Kiss and Tell" to find
5        the actress who portrayed Corliss Archer
6      task_type: Tool call
7      tool_name: wikipedia_search
8      input_parameters:
9        - name: query
10         type: string
11         value: Kiss and Tell Corliss Archer
12         is_reference: false
13     expected_output_parameters:
14       - name: actress_name
15         description: Name of the actress who portrayed Corliss Archer
16                      in Kiss and Tell
17         type: string
18     dependencies: []
19     execution_status: pending
20     execution_result: {}
21
22    - task_id: T2
23      task_description: |-
24        Search for the actress's career information
25      task_type: Tool call
26      tool_name: wikipedia_search
27      input_parameters:
28        - name: query
29          type: string
30          value: <JSON_PATH>T1.actress_name</JSON_PATH>
31          is_reference: true
32      expected_output_parameters:
33        - name: government_position
34          description: Government position held by the actress
35          type: string
36      dependencies: [T1]
37      execution_status: pending
38      execution_result: {}
39
40    - task_id: T3
41      task_description: |-
42        Provide the final answer stating the government position
43      task_type: Reasoning
44      tool_name: ""
45      input_parameters:
46        - name: position
47          type: string
48          value: <JSON_PATH>T2.government_position</JSON_PATH>
49          is_reference: true
50      expected_output_parameters:
51        - name: final_answer
52          description: The government position held by the woman
53          type: string
54      dependencies: [T2]
55      execution_status: pending
56      execution_result: {}
```

**Global Memory State:**

```
1  {}  # Empty - no tasks executed yet
```

## C.3  STEP 2: EXECUTE TASK T1 (TOOL CALL)

**Task Execution:**

*Tool:* wikipedia_search

*Resolved Input Parameters:*

```
1  query: "Kiss and Tell Corliss Archer"
```

*Tool Response (Truncated):*

```
Kiss and Tell is a 1945 American comedy film starring Shirley Temple
as Corliss Archer. The film was directed by Richard Wallace and
produced by Columbia Pictures. In her role as Corliss Archer, Temple
portrays a mischievous teenager who becomes involved in a series of
comedic misunderstandings. The character of Corliss Archer was
originally created by F. Hugh Herbert...

[Additional content about the film plot, production, and reception]
```

**Entity Extractor Invocation:**

*Input:*

```
1   api_response: |
2     Kiss and Tell is a 1945 American comedy film starring Shirley Temple
3     as Corliss Archer. The film was directed by Richard Wallace...
4
5   api_description: |-
6     Search for information about the film "Kiss and Tell" to find the
7     actress who portrayed Corliss Archer
8
9   entities:
10    - name: actress_name
11      description: Name of the actress who portrayed Corliss Archer
12                   in Kiss and Tell
13      type: string
```

*Entity Extractor Output:*

```
1  confidence_score: 0.95
2  extracted_entities:
3    actress_name: "Shirley Temple"
4  entities_summary: |-
5    The Wikipedia article explicitly states that "Kiss and Tell is a
6    1945 American comedy film starring Shirley Temple as Corliss Archer."
7    The actress name is clearly identified as Shirley Temple with very
8    high confidence (0.95).
```

**KPI Evaluation:**

- Confidence score: $0.95 \geq 0.7$
- All required entities extracted: $\text{actress\_name} \neq \text{null}$
- **Result:** KPI SUCCESS

**Memory Update:**

```
1  {
2    "T1": {
3      "actress_name": "Shirley Temple"
4    }
5  }
```

**Task Status Update:**

```
1  T1:
2    execution_status: done
3    execution_result:
4      actress_name: "Shirley Temple"
```

## C.4  STEP 3: EXECUTE TASK T2 (TOOL CALL) - INITIAL ATTEMPT

**Task Execution:**

*Tool:* wikipedia_search

*Input Parameter Resolution:*

```
Original: <JSON_PATH>T1.actress_name</JSON_PATH>
Memory lookup: T1.actress_name -> "Shirley Temple"
Resolved: "Shirley Temple"
```

*Resolved Input Parameters:*

```
1  query: "Shirley Temple"
```

*Tool Response (Truncated):*

```
Shirley Temple Black (April 23, 1928 - February 10, 2014) was an
American actress, singer, dancer, and diplomat. She was Hollywood's
number one box-office draw as a child actress from 1934 to 1938.

Early Life and Career:
Born in Santa Monica, California, Temple began her film career at
the age of three. She achieved unprecedented popularity in the 1930s,
appearing in films such as Bright Eyes (1934), Curly Top (1935), and
The Little Colonel (1935)...

Personal Life:
Temple married John Agar in 1945, but they divorced in 1950. She
then married Charles Alden Black in 1950, taking the name Shirley
Temple Black...

[Response focuses heavily on her acting career and personal life,
with minimal mention of her later diplomatic work]
```

**Entity Extractor Invocation:**

*Input:*

```
1  api_response: |
2    Shirley Temple Black (April 23, 1928 - February 10, 2014) was an
3    American actress, singer, dancer, and diplomat...
4    [Full response focused on acting career]
5
6  api_description: |-
7    Search for the actress's career information
8
9  entities:
10   - name: government_position
11     description: Government position held by the actress
12     type: string
```

*Entity Extractor Output:*

```
1  confidence_score: 0.15
2  extracted_entities:
3    government_position: null
4  entities_summary: |-
5    The Wikipedia response mentions that Shirley Temple was a "diplomat"
6    but does not specify any particular government position or title.
7    The article focuses primarily on her acting career and personal life.
8    Without a specific government position mentioned, the entity cannot
9    be extracted. Confidence score is very low (0.15) due to lack of
10   specific information.
```

**KPI Evaluation:**

- Confidence score: $0.15 < 0.7$
- Required entity: government_position = null
- **Result:** KPI FAILURE

**Memory State (Unchanged):**

```
{
  "T1": {
    "actress_name": "Shirley Temple"
  }
  // T2 failed - no entities stored
}
```

**Task Status Update:**

```
T2:
  execution_status: failed
  execution_result: {}
  failure_reason: "Entity 'government_position' not found in tool
                  response. Confidence very_low (0.2)."
```

## C.5 Step 4: Re-planning Triggered

**Re-planner Agent Invocation:**

*Input:*

```
tools:
  - name: wikipedia_search
    description: Search Wikipedia for information
    parameters:
      - name: query
        type: string
        required: true

query: "What government position was held by the woman who portrayed
        Corliss Archer in the film Kiss and Tell?"

current_plan:
  tasks:
    - task_id: T1
      execution_status: done
      execution_result:
        actress_name: "Shirley Temple"
    - task_id: T2
      execution_status: failed
      execution_result: {}
    - task_id: T3
      execution_status: pending

failed_task_id: T2

feedback: |-
  Task T2 failed during entity extraction. The tool response mentioned
  that Shirley Temple was a "diplomat" but did not specify any
  particular government position or title. The article focused
  primarily on her acting career and personal life without detailing
  her diplomatic roles. Entity 'government_position' could not be
  extracted (null value, confidence very_low = 0.2).
```

*Re-planner Output (Continuation Plan):*

```
tasks:
  - task_id: T2a
    task_description: |-
      Search specifically for Shirley Temple's diplomatic career and
      government positions with more targeted query
    task_type: Tool call
    tool_name: wikipedia_search
    input_parameters:
      - name: query
        type: string
        value: <JSON_PATH>T1.actress_name</JSON_PATH> diplomat
              government position Chief of Protocol Ambassador
```

```
13        is_reference: true
14    expected_output_parameters:
15      - name: government_position
16        description: |-
17          Specific government position title held by Shirley Temple,
18          such as Ambassador, Chief of Protocol, or other official
19          diplomatic role. Extract the exact position title, not
20          just "diplomat".
21        type: string
22    dependencies: [T1]
23    execution_status: pending
24    execution_result: {}
25
26  - task_id: T3a
27    task_description: |-
28      Provide the final answer stating the government position
29    task_type: Reasoning
30    tool_name: ""
31    input_parameters:
32      - name: position
33        type: string
34        value: <JSON_PATH>T2a.government_position</JSON_PATH>
35        is_reference: true
36    expected_output_parameters:
37      - name: final_answer
38        description: The government position held by the woman
39        type: string
40    dependencies: [T2a]
41    execution_status: pending
42    execution_result: {}
```

**Plan Merge:** The continuation plan replaces T2 and T3 with T2a and T3a. The system now proceeds with the updated plan.

**Memory State (Still Unchanged):**

```
1  {
2    "T1": {
3      "actress_name": "Shirley Temple"
4    }
5    // T2 failed - still no new entities
6  }
```

## C.6 Step 5: Execute Task T2a (Tool Call) - RETRY WITH IMPROVED QUERY

**Task Execution:**

*Tool:* wikipedia_search

*Input Parameter Resolution:*

```
Original: <JSON_PATH>T1.actress_name</JSON_PATH> diplomat government
          position Chief of Protocol Ambassador
Memory lookup: T1.actress_name -> "Shirley Temple"
Resolved: "Shirley Temple diplomat government position Chief of
           Protocol Ambassador"
```

*Resolved Input Parameters:*

```
1  query: "Shirley Temple diplomat government position Chief of Protocol
2         Ambassador"
```

*Tool Response (Truncated):*

```
Shirley Temple Black (April 23, 1928 - February 10, 2014) was an
American actress, singer, dancer, and diplomat.

Diplomatic Career:
After her retirement from film and a brief television career, Temple
became actively involved in the Republican Party. Her diplomatic
appointments included:

- United States Ambassador to Ghana (1974-1976): Appointed by
```

```
President Gerald Ford, she served as the U.S. representative to
Ghana during a critical period in African-American relations.

- Chief of Protocol of the United States (1976-1977): In this role,
  Temple was responsible for advising the President, Vice President,
  and Secretary of State on matters of diplomatic procedure,
  organizing state visits, and managing protocol for foreign dignitaries.

- United States Ambassador to Czechoslovakia (1989-1992): Appointed
  by President George H. W. Bush, she served during the Velvet
  Revolution and the fall of communism in Eastern Europe.

Temple was the first woman to serve as Chief of Protocol and received
numerous awards for her diplomatic service...
```

**Entity Extractor Invocation:**

*Input:*

```
1  api_response: |
2    Shirley Temple Black (April 23, 1928 - February 10, 2014)...
3    [Full diplomatic career section as above]
4
5  api_description: |-
6    Search specifically for Shirley Temple's diplomatic career and
7    government positions with more targeted query
8
9  entities:
10   - name: government_position
11     description: |-
12       Specific government position title held by Shirley Temple,
13       such as Ambassador, Chief of Protocol, or other official
14       diplomatic role. Extract the exact position title, not
15       just "diplomat".
16     type: string
```

*Entity Extractor Output:*

```
1  confidence_score: 0.98
2  extracted_entities:
3    government_position: "Chief of Protocol"
4  entities_summary: |-
5    The Wikipedia response explicitly lists multiple government positions
6    held by Shirley Temple. The article clearly states "Chief of Protocol
7    of the United States (1976-1977)" as one of her positions, along with
8    detailed description of the role. This is extracted with very high
9    confidence (0.98) as it is explicitly stated with the exact title
10   format requested.
```

**KPI Evaluation:**

- Confidence score: $0.98 \geq 0.7$
- All required entities extracted: government_position $\neq$ null
- **Result:** KPI SUCCESS

**Memory Update:**

```
1  {
2    "T1": {
3      "actress_name": "Shirley Temple"
4    },
5    "T2a": {
6      "government_position": "Chief of Protocol"
7    }
8  }
```

**Task Status Update:**

```
1  T2a:
2    execution_status: done
3    execution_result:
4      government_position: "Chief of Protocol"
```

## C.7 Step 6: Execute Task T3a (Reasoning)

**Task Execution:**

*Input Parameter Resolution:*

```
Original: <JSON_PATH>T2a.government_position</JSON_PATH>
Memory lookup: T2a.government_position -> "Chief of Protocol"
Resolved: "Chief of Protocol"
```

**Reasoning Agent Invocation:**

*Input:*

```
1  task_description: |-
2    Provide the final answer stating the government position
3  input_parameters:
4    - name: position
5      type: string
6      value: "Chief of Protocol"
7  expected_output_parameters:
8    - name: final_answer
9      description: The government position held by the woman
10     type: string
```

*Reasoning Agent Output:*

```
1  Justifications:
2  1. The original question asks for the government position held by
3     the woman who portrayed Corliss Archer in "Kiss and Tell"
4  2. From previous tasks, we identified the actress as Shirley Temple
5  3. The input parameter provides the government position as
6     "Chief of Protocol"
7  4. This is the answer to the original question
8
9  execution_result:
10   status: completed
11   outputs:
12     final_answer: "Chief of Protocol"
13 execution_details:
14   reasoning_steps:
15     - "Retrieved government position from input parameters"
16     - "Verified this answers the original question"
17     - "Formatted as final answer"
```

**Validation:**

- Status: completed
- All expected outputs present: final_answer $\neq$ null
- **Result:** Task SUCCESS

**Final Memory Update:**

```
1  {
2    "T1": {
3      "actress_name": "Shirley Temple"
4    },
5    "T2a": {
6      "government_position": "Chief of Protocol"
7    },
8    "T3a": {
9      "final_answer": "Chief of Protocol"
10   }
11 }
```

**Task Status Update:**

```
1  T3a:
2    execution_status: done
3    execution_result:
4      final_answer: "Chief of Protocol"
```

## C.8 Execution Summary

**Final Answer:** Chief of Protocol

**Execution Statistics:**

- Total tasks executed: 4 (T1, T2-failed, T2a, T3a)
- Successful tasks: 3 (T1, T2a, T3a)
- Failed tasks: 1 (T2)
- Tool calls: 3 (T1, T2, T2a)
- Reasoning tasks: 1 (T3a)
- Entity extraction invocations: 3 (for T1, T2, T2a)
- Replanning events: 1 (after T2 failure)
- Total LLM calls: 8
    - 1 initial planner
    - 2 tool executions (T1, T2)
    - 2 entity extractions (T1, T2)
    - 1 re-planner
    - 1 tool execution (T2a)
    - 1 entity extraction (T2a)
    - 1 reasoning (T3a)
- Execution status: SUCCESS (after recovery)

**Memory Evolution Timeline:**

Table 2: Global Memory State Evolution

| Step | Event | Memory State |
|---|---|---|
| 0 | Initial | `{}` |
| 1 | T1 success | `{"T1": {"actress_name": "Shirley Temple"}}` |
| 2 | T2 failure | `{"T1": {"actress_name": "Shirley Temple"}}` |
| 3 | Re-planning | `{"T1": {"actress_name": "Shirley Temple"}}` |
| 4 | T2a success | `{"T1": {"actress_name": "Shirley Temple"}, "T2a": {"government_position": "Chief of Protocol"}}` |
| 5 | T3a success | `{"T1": {"actress_name": "Shirley Temple"}, "T2a": {"government_position": "Chief of Protocol"}, "T3a": {"final_answer": "Chief of Protocol"}}` |

**Note:** For clarity and readability, the memory states shown in the timeline above display only the extracted entities component. In practice, the global memory also maintains the `current_plan` structure with execution statuses, task descriptions, and results for each task. This dual structure enables: (1) efficient parameter binding via entity references, and (2) comprehensive execution context for the re-planner when generating continuation plans. The re-planner accesses the `current_plan` to understand which tasks succeeded (T1), which failed (T2), and what entities are available in memory, allowing it to generate T2a that reuses T1's output while addressing T2's specific failure.

**Task Dependency Chain with Re-planning:**

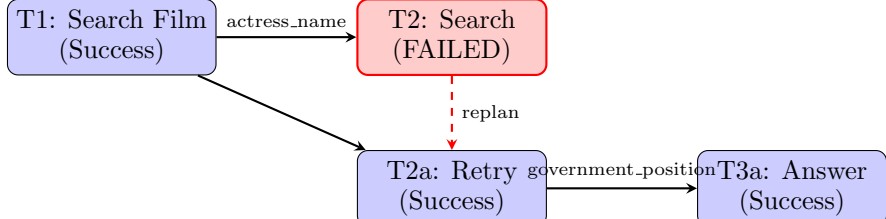

**Key Observations:**

- **Failure Detection:** The entity extractor correctly identified that the initial search for "Shirley Temple" did not provide specific government position information, returning null with very low confidence (0.2).

- **Memory Preservation:** The successfully extracted entity from T1 (`actress_name = "Shirley Temple"`) was preserved in global memory throughout the failure and re-planning process.

- **Intelligent Re-planning:** The re-planner analyzed the failure feedback and generated a more targeted search query including keywords like "diplomat", "government position", "Chief of Protocol", and "Ambassador" to improve retrieval.

- **Enhanced Entity Description:** The re-planner improved the entity description in T2a to be more specific: "Extract the exact position title, not just 'diplomat'" which guided the entity extractor to look for precise government position names.

- **Continuation-Based Recovery:** Rather than restarting from scratch, the system continued from the failure point (T2) using T2a, reusing the valid `actress_name` entity from T1's memory.

- **Successful Recovery:** After re-planning with a more specific query, T2a successfully retrieved detailed diplomatic career information and extracted "Chief of Protocol" with very high confidence (1.0).

- **Efficient Parameter Flow:** JSON-path references (`<JSON_PATH>T1.actress_name</JSON_PATH>` and `<JSON_PATH>T2a.government_position</JSON_PATH>`) enabled clean data flow between tasks using the global memory system.

- **Final Answer Accuracy:** The framework successfully recovered from the initial failure and produced the correct answer "Chief of Protocol".

**Comparison to Baseline Approaches:**

- **ReAct:** Would likely continue with vague "diplomat" information without systematic validation, potentially producing an incomplete or incorrect answer.

- **Plan-and-Execute:** Would fail at T2 without structured entity validation and likely restart from scratch rather than preserving T1's successful result, wasting computational resources.

- **KPI-Chain:** Detected the specific failure (missing government position), preserved successful work (actress name), and intelligently adapted the search strategy to succeed on retry.

