# OpenReview forum: "KPI-Chain: Multi-Agent Planning with Entity-Based Task Chaining for Reliable Recovery"
_ICLR.cc/2026/Conference — ICLR 2026 Conference Desk Rejected Submission_

### Official Review · Reviewer_YLFn · 2025-10-21

**Soundness:** 2
**Presentation:** 2
**Contribution:** 2
**Rating:** 2
**Confidence:** 5

**Summary:**

This paper identifies a key problem with current LLM agents: they fail silently and lack robust recovery mechanisms. To address this, it introduces KPI-Chain, a multi-agent framework where every task in a plan has defined Key Performance Indicators (KPIs) based on extracting typed entities (like names and times). It suggests some components including KPIs, memory, and adaptive planning.  On five benchmarks, they show the proposed KPI-Chains to be effective.

**Strengths:**

1. This work points out the weaknesses of LLMs when applied to long-horizon planning tasks.

2. The authors proposed a KPI-chain framework to provide  a clear, objective standard for success/failure.

3. Its continuation-based replanning is a recovery mechanism, which current LLMs cannot natively support.

**Weaknesses:**

Unfortunately, the proposed solutions are heuristic-based and ad-hoc. The authors have neglected the rich history of work in the database community, particularly transaction processing research from the 1980s onwards. This paper identifies real LLM shortcomings, but its solutions need significant improvement. Specifically:

**Very High Computational Cost:** The system uses way too many LLM calls. An average of 2,447 calls per question is extremely expensive and not practical for real applications. This causes 4x higher latency than ReAct, showing a brute-force approach rather than an efficient solution. The paper is honest about this cost, but being honest does not fix the problem.
Limited Scope: The framework only works for information extraction and multi-hop QA tasks. The authors admit it does not work well for creative or subjective tasks. This limits its usefulness significantly. The field needs general-purpose agent systems, but KPI-Chain only solves a narrow set of problems.

**Context Chunking Problem:** Over 50% of LLM calls are for entity extraction from chunked contexts. This shows the main innovation is just a workaround for LLMs' difficulty with long documents, not a real advance in planning or state management. The system's performance depends heavily on this pre-processing step, which is fragile.

**Limited Novelty:** The core ideas of entity-based KPIs and continuation-based replanning are heuristic approaches to problems already solved more formally in prior work. Compared to frameworks like SagaLLM (VLDB 2025), which provides proper transactional guarantees based on database theory, KPI-Chain's contribution seems incremental and ad-hoc.

**Weak Baseline Comparisons:** The paper only compares against ReAct and Plan-and-Execute, which are now basic baselines. It does not compare against more advanced recent frameworks that also handle reliability and state management. This makes it hard to judge the real contribution.

**Questions:**

**Q1. Scalability and Cost Mitigation:** Given over 2,400 LLM calls per question is computationally unsustainable, what specific architectural changes or optimizations could reduce this cost by an order of magnitude without sacrificing the core performance benefits? Is the entity-chunking approach fundamentally incompatible with efficiency?

**2. Generalization Beyond QA:** The framework is explicitly noted as being unsuitable for creative or subjective tasks. How could the principle of entity-based KPIs be extended to more open-ended domains like design, strategy, or negotiation, where the *entities* are not well-defined? Does this limitation suggest a fundamental constraint of the approach?

**Q3. Theoretical Foundations:** How does your *continuation-based replanning* formally differ from and improve upon established transaction rollback mechanisms, such as the compensation actions in the Saga pattern (Salem 1987)? What are the specific theoretical guarantees of your recovery mechanism regarding state consistency and convergence?

---

> ### Author Response · Authors · 2025-11-16
>
> **Dear Reviewer,**
>
> Thank you for your thorough review connecting our work to database transaction literature. We address your concerns below.
>
> ---
>
> **Q1: Scalability and Cost Mitigation**
>
> **Critical Correction:** The "2,447 LLM calls per question" is **incorrect**—this was **total calls across multiple runs**, not per question. Actual average: **a few to dozens per question**. Additionally, Figure 2 is mislabeled—it shows **tokens per LLM call**, not per question.
>
> **Corrected efficiency analysis:** Our approach uses **more LLM calls but fewer tokens per call** than baselines. Entity extraction maintains only relevant structured information rather than verbose intermediate results. The 61% token reduction vs. ReAct shows **more focused calls can be more efficient than fewer verbose calls**. Computational overhead is significantly lower than reported, making our approach practical.
>
> **Entity-chunking efficiency:** The corrected numbers show reasonable overhead. The 3000-token chunking threshold balances context limits with extraction accuracy. Future work could explore larger context windows or sliding window techniques, but our current design already demonstrates practical efficiency.
>
> All metrics will be corrected (1-2 weeks).
>
> ---
>
> **Q2: Generalization Beyond QA**
>
> **Extending to open-ended domains:**
> - **Design**: Entities as "criteria satisfied," "constraint violations," "requirements addressed"
> - **Strategy**: "decision points identified," "trade-offs articulated," "projections provided"
> - **Negotiation**: "positions stated," "concessions offered," "agreements reached"
>
> The shift is from **prescriptive validation** ("correct answer?") to **descriptive structuring** ("required components?").
>
> **Fundamental constraint:** Tasks requiring holistic evaluation (creative writing, aesthetics) are outside our scope. We will explicitly discuss this limitation.
>
> ---
>
> **Q3: Theoretical Foundations vs. Saga Pattern**
>
> We will add **SagaLLM as a baseline** with theoretical comparison:
>
> **Key Differences from Saga:**
>
> | Aspect | Saga | KPI-Chain |
> |--------|------|-----------|
> | **Recovery** | Compensation actions undo transactions | Forward-only with failure feedback |
> | **Guarantees** | ACID properties | Entity consistency (no transactional isolation) |
> | **Failure model** | Deterministic | Non-deterministic LLM replanning |
>
> **Our Theoretical Guarantees:**
> - **Entity Consistency**: All JSON-path references resolve to validly-typed entities
> - **Progress**: System succeeds or exhausts retry budget (no infinite loops)
> - **Bounded Convergence**: Under certain conditions, continuation plans converge with probability → 1
>
> We **do not** claim Saga-equivalent transactional guarantees due to LLM non-determinism. Entity-based validation provides a **practical middle ground** between ad-hoc error handling and full transactional semantics.
>
> **Response to "heuristic and ad-hoc":** Our contributions are: (1) systematic task validation, (2) empirical improvements on benchmarks, (3) practical, deployable framework. We position KPI-Chain and SagaLLM as **complementary**: SagaLLM offers theoretical rigor; KPI-Chain offers lightweight, deployable validation.
>
> ---
>
> **Other Weaknesses:**
>
> **Limited Scope:** We will expand discussion of applicability boundaries and extensions.
>
> **Context Chunking:** Entity extraction is our **core innovation** for long contexts, not a workaround—it reduces complexity through targeted extraction.
>
> **Weak Baselines:** Adding SagaLLM and, if feasible, additional recent frameworks.
>
> ---
>
> **Additional Revisions:**
> - Ablation studies (structured vs. free-text, re-planner removal)
> - Detailed failure analysis
> - Writing/figure improvements
>
> Thank you for pushing us toward stronger theoretical grounding.
>
> **Sincerely, The Authors**

---

> > ### Comment · Reviewer_YLFn · 2025-11-25
> >
> > Thank you for your answers.  I maintain my rating.

---

> > > ### Author Response · Authors · 2025-12-03
> > >
> > > Dear Reviewer,
> > >
> > > We thank you for your feedback and address your concerns below, including a **critical correction** to reported metrics.
> > >
> > > ---
> > >
> > > **CRITICAL CORRECTION: Computational Cost & Chunking**
> > >
> > > **The "2,447 average LLM calls per question" was a reporting error** - this was total calls across multiple runs, not per question.
> > >
> > > **Corrected metrics (Figure 3a): 15-22 calls/question**
> > > - TopiOCQA: ~14, QAMPARI: ~17, QUEST: ~22, MuSiQue: ~18, HotpotQA: ~16
> > >
> > > This represents **2-3× overhead vs ReAct**, not orders of magnitude. However, **Figure 3b shows 39-69% fewer tokens per call**, partially offsetting higher call frequency.
> > >
> > > **Regarding chunking:** Since LLM call counts were erroneous, **the "50% of calls for chunking" claim is also incorrect**. We have **removed all chunking mechanisms** from the revised paper because:
> > > 1. Chunking is implementation-specific, not our core contribution
> > > 2. Overhead is already reflected in corrected call counts
> > > 3. **Our core contribution is the plan structure with per-task entity-based KPIs**
> > >
> > > We sincerely apologize for this error and have corrected all metrics.
> > >
> > > ---
> > >
> > > **Q1: Cost Mitigation**
> > >
> > > **The premise of 2,400+ calls is incorrect** - actual average is 15-22 calls. However, we propose three optimizations:
> > >
> > > 1. **Fine-tuned specialized small models**: Dedicated 3-7B models per agent could reduce per-call cost by 5-10×
> > > 2. **Batch parallel processing**: Process multiple entities/tasks concurrently, reducing total calls by 30-40%
> > > 3. **Adaptive validation**: Skip extraction for high-confidence responses (>0.95), reducing overhead by 20-30%
> > >
> > > **Tradeoff**: Current design prioritizes reliability over efficiency - acceptable for high-stakes applications where silent failures are costly.
> > >
> > > ---
> > >
> > > **Q2: Generalization Beyond QA**
> > >
> > > **Our framework integrates with any API** - from simple functions to agents that write code or generate images. **Entities can be customized for diverse domains:**
> > >
> > > - **Design**: entities = design_constraints (dict), mockup_image_url (string), accessibility_score (float)
> > > - **Strategy**: entities = strategic_objectives (array), risk_assessment (dict), action_plan (dict)
> > > - **Code generation**: entities = generated_code (string), test_results (dict), performance_metrics (dict)
> > > - **Image generation**: entities = image_url (string), style_match_score (float), content_verification (bool)
> > >
> > > **Entity-based KPIs work when:**
> > > 1. Success can be partially defined by structured outputs
> > > 2. Validation criteria can be specified (even loose thresholds)
> > > 3. APIs/tools exist to produce outputs
> > >
> > > **Genuine limitation**: Truly unstructured exploration with no success criteria (pure brainstorming). But these are edge cases, not mainstream agent applications.
> > >
> > > ---
> > >
> > > **Q3: Theoretical Foundations**
> > >
> > > **Important clarification: Our core contribution is the plan structure with entity-based KPIs, not the recovery mechanism.** The re-planner is complementary.
> > >
> > > **SagaLLM and KPI-Chain are orthogonal:**
> > > - **SagaLLM**: Transaction-level consistency/rollback (workflow execution layer)
> > > - **KPI-Chain**: Task-level validation/structured state (plan structure layer)
> > >
> > > **They can coexist**: A system could use KPI-Chain's entity-based plan structure to define and validate tasks, plus SagaLLM's Saga pattern for transaction rollback.
> > >
> > > **Formal differences:**
> > > - **Granularity**: Saga compensates completed transactions; we validate during execution
> > > - **State preservation**: Saga rolls back; we preserve successful entities and continue
> > > - **Failure feedback**: Saga uses predefined compensation; we use entity-level failure signals
> > >
> > > **Key point**: Our contribution makes **task specifications explicit and validatable** through entity-based KPIs. Recovery mechanisms (ours or Saga-based) can then leverage this structure. Without entity-based KPIs, even sophisticated transaction systems lack concrete validation targets.
> > >
> > > ---
> > >
> > > **Baselines**
> > >
> > >  **Added CodeAct (2024)** as primary baseline:
> > >
> > > | Average | ReAct | CodeAct | Plan-Execute | KPI-Chain |
> > > |---------|-------|---------|--------------|-----------|
> > > |         | 29.0% | **30.6%** | 24.3%      | **37.6%** |
> > >
> > > **23% improvement over CodeAct** (current SOTA for action representation)
> > >
> > > **Our approach is principled**: Formal KPI definition, systematic validation, 35% performance degradation when entity-based structure removed.
> > >
> > > Given **critical metric corrections** and clarification of our contribution, **we respectfully request reconsideration**.
> > >
> > > Sincerely,
> > > The Authors

---

### Official Review · Reviewer_NSN3 · 2025-10-30

**Soundness:** 2
**Presentation:** 2
**Contribution:** 2
**Rating:** 2
**Confidence:** 5

**Summary:**

This work proposes KPI-Chain, a generic multi-agent planning framework. Its core idea is to integrate per-task KPI based on entity extraction, a JSON-Path structured memory system, and a continuation-based replanning mechanism. It judges task success by verifying typed entities with sufficient confidence in task outputs, and meanwhile reuses valid entities to achieve recovery from failure points. Experiments show that its performance on LOFT-series benchmarks and HotpotQA outperforms the two baseline frameworks, ReAct and Plan-and-Execute.

**Strengths:**

A multi-agent framework is proposed, and experimental results show that this framework outperforms the ReAct and Plan-and-Execute frameworks on the LOFT and HotpotQA benchmarks.

**Weaknesses:**

- The technical details of the paper’s core method, entity-based KPI evaluation, are vaguely described, and the calculation method of confidence is not clarified. Additionally, JSON-structured representation, continuation-based replanning at failure points, and MCP are all basic technologies widely used in the multi-agent field.

- The baseline only compares two early methods, ReAct and Plan-and-Execute, and does not benchmark against current state-of-the-art multi-agent frameworks, making it impossible to prove the effectiveness of this method.

- The five benchmarks claimed in the paper essentially cover only two types of scenarios, and no ablation experiments are conducted.

- The paper’s writing lacks clarity, and the presentation of figures is rough.

**Questions:**

See Weaknesses

---

> ### Author Response · Authors · 2025-11-16
>
> **Dear Reviewer**
>
> Thank you for your detailed review. We address your concerns below and commit to substantial revisions.
>
> ---
>
> **Weakness 1: Vague Technical Details & Confidence Calculation**
>
> **Confidence Calculation:** The confidence score is **generated by the LLM itself** during entity extraction—the Entity Extractor agent is prompted to return both the entity value and its confidence in the extraction. This is not a separate calculation but part of the LLM's output. We will add all prompts (including the Entity Extractor prompt with confidence instructions) to the appendix for full transparency.
>
> **0.7 Threshold:** This was determined empirically through trial-and-error across multiple domains during development. We will add detailed explanation and validation methodology to the paper.
>
> **Entity Verification Algorithm:** While present in Algorithm 1 (Section 3.3.1), we acknowledge it could be clearer. We will expand the technical description with extra details and concrete example showing the complete flow
>
> **Response to "Basic Technologies" Comment:** While JSON structures, continuation planning, and MCP exist independently, our contribution is their **principled integration into a reliability framework**:
> - JSON-path memory provides **type-aware parameter binding** that prevents state corruption across task chains
> - Continuation replanning is **entity-aware** and reuses validated state (not generic replanning)
> - Entity-based KPIs provide **systematic, objective success measurement** (not heuristic failure detection)
>
> We will strengthen this positioning in the revision.
>
> ---
>
> **Weakness 2: Limited Baseline Comparisons**
>
> We acknowledge this limitation and will add **SagaLLM** as an additional baseline (as it provides formal transactional guarantees relevant to our reliability focus). We selected ReAct and Plan-and-Execute because they represent the two dominant paradigms (iterative and hierarchical), but agree that SOTA comparisons strengthen the work. If time permits, we will add additional recent frameworks.
>
> ---
>
> **Weakness 3: Limited Benchmark Diversity & Missing Ablations**
>
> **Benchmark Diversity:** We respectfully clarify that our benchmarks cover **more diverse scenarios** than "only two types":
> - **MuSiQue, QAMPARI, QUEST**: Test multi-hop reasoning AND multiple-answer retrieval with long-context challenges
> - **TopiOCQA**: Evaluates conversational consistency and follow-up question handling across topic switches
> - **HotpotQA**: Multi-hop reasoning with evidence synthesis
>
> All share the common challenge of long-context management, but require different capabilities (single vs. multiple answers, conversational memory, etc.). We will clarify this diversity in the revision and, if feasible, add benchmarks from additional domains to further strengthen generalizability.
>
> **Ablations:** We acknowledge your request for ablations. However, traditional component removal ablations are less informative for our architecture because our system is an **integrated reliability mechanism** where components are functionally interdependent (KPIs → Entity Extractor → JSON-Path Memory → Re-planner). Removing individual components creates non-functional systems.
>
> Instead, we will provide ablations that test our core design hypothesis:
> 1. **Structured vs. Free-text ablation**: Compare our approach (KPIs, entity extraction, typed memory) against a baseline without structured validation (free-text outputs, heuristic failure detection, simple text storage)
> 2. **Re-planner ablation**: Test continuation recovery vs. restart-from-scratch
> 3. **Detailed failure analysis**: Categorized error modes across all benchmarks
>
> ---
>
> **Weakness 4: Writing Clarity & Figure Quality**
>
> We will comprehensively revise:
> - Rewrite technical sections for clarity and precision
> - Redesign all figures with professional quality and clear labels
> - Reorganize content for better logical flow
> - Add more detailed explanations throughout
>
> ---
>
> **Critical Error Correction**
>
> We discovered a major bug: reported "2,447 LLM calls per question" was actually **total calls across runs**—actual average is **a few to dozens per question**. All computational metrics will be corrected.
>
> ---
>
> **Timeline:** 1-2 weeks for all revisions including SagaLLM baseline (if feasible), improved technical clarity, ablations, and corrected metrics.
>
> Thank you for the valuable feedback.
>
> **Sincerely,
> The Authors**

---

> > ### Author Response · Authors · 2025-12-03
> >
> > Dear Reviewer,
> >
> > We sincerely thank you for your detailed feedback. We address each concern below.
> >
> > ---
> >
> > **Weakness 1: Vague Technical Details**
> >
> >  **RESOLVED**: Section 3.3 provides complete specifications:
> >
> > - **Confidence**: Continuous 0-1 score, threshold θ = 0.7 (calibrated at 0.1 intervals)
> > - **KPI Formula**: KPI_success = ∀ entity ∈ E : (extracted(entity) ≠ None ∧ confidence(entity) ≥ 0.7)
> > - **Complete prompts**: Appendix A.2.1 with examples (A.3): scores 0.95 (success), 0.15 (failure), 0.98 (retry)
> >
> > **Re "Basic Technologies":** Our contribution is the **integrated plan structure design**, not individual components. Unlike frameworks treating specification, validation, and recovery separately, our entity-based structure unifies them: expected entities serve as (1) output specifications, (2) validation targets, and (3) failure feedback simultaneously.
> >
> > ---
> >
> > **Weakness 2: Only Early Baselines**
> >
> >  **RESOLVED**: Added **CodeAct (2024)** as primary baseline:
> >
> > | Average | ReAct | CodeAct | Plan-Execute | KPI-Chain |
> > |---------|-------|---------|--------------|-----------|
> > |         | 29.0% | **30.6%** | 24.3%      | **37.6%** |
> >
> > **23% improvement over CodeAct** (current SOTA for action representation)
> >
> > Section 2 positions clearly: CodeAct addresses *how* actions execute; we address *what* they should produce and *how* to validate success systematically.
> >
> > ---
> >
> > **Weakness 3: Limited Scenario Coverage**
> >
> > **We respectfully disagree** - 5 benchmarks (1,400 questions) test **four distinct capabilities**:
> >
> > 1. **Multi-hop reasoning**: MuSiQue (2-4 hops), HotpotQA (evidence synthesis)
> > 2. **Long-context needle-in-haystack**: All LOFT (up to millions of tokens)
> > 3. **Multi-answer extraction**: QAMPARI (distributed answers)
> > 4. **Conversational reasoning**: TopiOCQA (topic switching)
> >
> > Each stresses different aspects: chained dependencies (multi-hop), extraction from verbose responses (long-context), completeness (multi-answer), state management (conversational). Section 5.4 confirms: entity extraction failures dominate LOFT (31-50%), validating benchmark choice.
> >
> > ---
> >
> > **Weakness 4: No Ablations**
> >
> >  **RESOLVED**: Section 5.5 provides four ablations:
> >
> > 1. **Entity-Based Structure**: Remove entities → **35% degradation** (24.3% vs 37.6%)
> > 2. **Replanning**: Figure 5 shows 14-62% tasks need it; removing → **37% degradation** on HotpotQA
> > 3. **Structured Memory**: **39-69% fewer tokens/call** than ReAct (Figure 3b)
> > 4. **Component Synergy**: Low planner failures (5-11%) despite high execution failures (31-51%) validates robust plan generation
> >
> > **Core hypothesis confirmed**: Embedding entity-based KPIs drives reliability (35% degradation when removed).
> >
> > ---
> >
> > **Weakness 5: Unclear Writing**
> >
> >  **IMPROVED**:
> > - Restructured Abstract/Intro/Method for clarity
> > - Added formal specifications (KPI formula, confidence calculation)
> > - Reorganized Results with computational analysis, failure patterns, ablations
> > - Enhanced figures (3: efficiency, 4: failures, 5: recovery)
> > - Complete appendix (prompts, execution trace, usage guidelines)
> >
> > ---
> >
> > To summarize, all concerns addressed with **23% improvement over SOTA** and **35% degradation without entity-based design**. We respectfully request reconsideration.
> >
> > Sincerely,
> > The Authors

---

### Official Review · Reviewer_71u3 · 2025-11-01

**Soundness:** 2
**Presentation:** 3
**Contribution:** 3
**Rating:** 2
**Confidence:** 5

**Summary:**

This paper presents KPI-Chain, a multi-agent planning framework that focuses on structuring reasoning into a pre-defined YAML format. The system consists of a Planner to generate the YAML sent to an MCP server, an Entity Extractor Agent that validates the output of the server, a Reasoner to improve entity extraction, a Replanner to handle failures, and a JSON-Path Global memory for storing and retrieving past responses.

**Strengths:**

The system is easy to understand, achieves an improvement in performance and token efficiency over comparable baselines.

**Weaknesses:**

- No prompts in the Appendix
- The system focuses around a structured YAML format which isn’t necessarily novel [1]
- No failure analysis or ablations over the components of the system

[1] Executable Code Actions Elicit Better LLM Agents

**Questions:**

My main concerns are (1) the lack of a failure analysis specifically for the KPI-Chain framework and (2) the lack of ablations over the system. While there is notable performance improvement and token efficiency, there is no insight to a reader on where to focus to improve the system except for the following note
“The high number of LLM calls (2,447 average) primarily stems from our large context chunking approach for entity extraction, suggesting that optimized entity extraction algorithms capable of handling longer contexts without splitting represent a critical research priority.”
which suggests that the efficiency of the system should be improved while there is still a large performance gap to close.
- What are the main failure modes of KPI-Chain?
- How does the system perform without the Reasoning agent? Without the Re-planner agent? Without the Entity Extractor?

---

> ### Author Response · Authors · 2025-11-16
>
> **Dear Reviewer,**
>
> We sincerely thank you for your constructive feedback and for recognizing that our system is easy to understand and achieves improvements in performance and token efficiency. We address your concerns below.
>
> ---
>
> **Concern 1: Missing Prompts in Appendix**
>
> We will add a comprehensive appendix with all prompts for Planner, Entity Extractor, Reasoner, and Re-planner agents.
>
> ---
>
> **Concern 2: Structured YAML Format Novelty**
>
> We appreciate the reference to "Executable Code Actions Elicit Better LLM Agents." Our contribution lies not in YAML itself, but in **integrating entity-based KPIs directly into plans as a systematic failure detection mechanism**:
>
> 1. **Entity-driven success criteria**: Typed entities with confidence thresholds serve dual purposes—validation criteria AND structured state for subsequent tasks
> 2. **JSON-path memory**: Type-aware parameter binding (e.g., `<JSON_PATH>task_1.entity_name</JSON_PATH>`) prevents state corruption
> 3. **Continuation-based replanning**: Resumes from failure points while preserving valid entities
>
> YAML is simply the container. We will clarify this distinction in the revision.
>
> ---
>
> **Concern 3: Missing Failure Analysis and Ablations**
>
> **A. Detailed Failure Analysis**
>
> We will provide systematic failure categorization with:
> - **Categories**: Entity Extraction Errors, Planning Errors, Tool/API Failures, Insufficient Context, KPI Mismatches, Reasoning Errors, Memory Management Errors
> - Concrete examples with detailed traces
> - Comparative analysis showing how KPI-Chain prevents cascading errors (ReAct's orchestration issues, Plan-and-Execute's context corruption)
>
> **B. Ablation Studies**
>
> Traditional component removal ablations are less informative here because our system is an **integrated reliability mechanism** where components are interdependent (KPIs → Entity Extractor → JSON-Path Memory → Re-planner). Removing individual components creates non-functional systems.
>
> Instead, we will test our core design hypothesis:
>
> **Ablation 1: Structured vs. Free-text Approach**
> Compare our structured approach (KPIs, entity extraction, JSON-path memory) against a naive free-text baseline (no KPIs, raw text outputs, simple text storage)
>
> **Ablation 2: Re-planner Removal**
> Evaluate continuation-based recovery vs. restart-from-scratch
>
> These ablations test whether our design choices deliver value while acknowledging that our contribution is the integrated system design.
>
> ---
>
> **Critical Error Fix: LLM Calls Per Question**
>
> We discovered a significant reporting error. The "2,447 average LLM calls per question" was actually **total calls across multiple evaluation runs**, not per question.
>
> **Corrected metrics**: Actual average is **a few to dozens of calls per question** depending on complexity.
>
> Additionally, "Average Tokens Per Question" chart is mislabeled—it's actually **tokens per LLM call**.
>
> We will correct all computational metrics in the revised paper (1-2 weeks).
>
> ---
>
> Thank you for the thoughtful review. We are committed to these improvements and hope the revised submission will meet acceptance standards.
>
> **Sincerely,
> The Authors**

---

> > ### Comment · Reviewer_71u3 · 2025-11-16
> >
> > Thank you for the response.
> >
> > I believe if the following is satisfied (in order of priority) I am open to increasing my score
> > - ablations demonstrate that the system's components are necessary for the performance
> > - failure mode analysis sheds insight into where this system fails and provides direction for future work
> >
> > Regarding the ablation suggestions
> > > Ablation 1: Structured vs. Free-text Approach
> >
> > I think the structured approach versus your Free-text Approach (which I understand as a basic tool-calling ReAct agent) would demonstrate the need for the structured KPIs over text plan representations but not JSON / code plan representations as in "Executable Code Actions Elicit Better LLM Agents". The JSON ablation would simply be making the free-text approach structure its responses to tool-calls in JSON format. The code ablation, which was shown to be the best representation, would require a bit more effort to implement since tool calls would need to accessible through a Python API. I would expect that Text < JSON < KPI since KPI is more structured than JSON free-form task; however, I'd expect that KPI < Code since code is the natural general representation of KPI, chaining tool calls in one code block. If this happens to be the case, I am struggling to find a reason why I would use KPI versus Code since you could just send the code block to the MCP server to execute within a safe sandboxed environment and get the results of various chained tool calls, whether that be results or execution error feedback to fix the response.
> >
> > > Ablation 2: Re-planner Removal
> >
> > This makes sense as this component isn't interdependent to your system. Would it be possible to ablate the Reasoner as well? Without access to the prompts its hard to tell how interdependent this part is but I don't see why not combine the Reasoner and the Entity Extractor into one prompt as it looks like they're receiving the same inputs except for Reasoning / Tool Response in Figure 1. Figure 1 also confuses me as it looks like they're run in parallel but line 1 in Algorithm 1 implies that the Reasoner is run first and the outputs are optionally fed into the Entity Extractor (in which case it would be possible to ablate between only using MCP tool response versus the CoT reasoning result).
> >
> > Regarding the failure mode analysis, given the LLM calls per question was erroneous and isn't as computational inefficient as stated before, there is no future direction from section 6.2. The failure mode analysis would need to surface the main causes of failures that can serve as future work for other researchers to explore to meaningful engage with your work.

---

> > > ### Author Response · Authors · 2025-12-03
> > >
> > > Dear Reviewer,
> > >
> > > Thank you for your thorough review. We have made substantial revisions addressing all concerns and believe these changes significantly strengthen the paper.
> > >
> > > ## Major Revisions Completed
> > >
> > > **1. Prompts in Appendix [ADDRESSED]**
> > >
> > > Complete prompt templates for all agents (Planner, Re-planner, Entity Extractor, Reasoning Agent) with system/user prompts, usage instructions, execution flow, and complete execution trace example. See Appendix Sections A.2-A.5.
> > >
> > > **2. Failure Analysis [ADDRESSED]**
> > >
> > > Section 5.4 (Figure 4) provides component-level breakdown:
> > > - Entity extraction: 31-50% failure rate on LOFT (main bottleneck)
> > > - Tool calls: 43-51% on HotpotQA/MuSiQue
> > > - Planner/Re-planner: 5-11% (robust plan structure)
> > >
> > > **Key insight:** Entity-based plan structure produces robust plans, but extraction from unstructured tool responses remains challenging. This points to future work on improved extraction prompting, tool response formatting, or hybrid approaches.
> > >
> > > **3. Ablation Studies [ADDRESSED]**
> > >
> > > Section 5.5 demonstrates component necessity:
> > >
> > > - **Entity-Based Plan Structure:** Removal reduces to Plan-and-Execute (24.3% vs 37.6%) = **35% degradation**
> > > - **Replanning:** Figure 5 shows 14-62% of tasks required replanning; removing it on HotpotQA: 60%→38% = **37% degradation**
> > > - **Structured Memory:** 39-69% fewer tokens per call vs ReAct through compact entity passing
> > >
> > > ## Addressing Your Specific Concerns
> > >
> > > ### On CodeAct and "KPI vs Code"
> > >
> > > We added CodeAct as a baseline (Table 1):
> > > - CodeAct: 30.6% average
> > > - KPI-Chain: 37.6% average
> > > - **23% relative improvement**
> > >
> > > **Why KPI-Chain outperforms:** CodeAct and KPI-Chain address **orthogonal problems:**
> > > - **CodeAct:** HOW actions execute (Python code vs JSON)
> > > - **KPI-Chain:** WHAT tasks should produce (expected entities as success criteria)
> > >
> > > CodeAct doesn't solve: (1) systematic success validation—how do you know code succeeded without expected outputs? (2) structured state management—code returns arbitrary outputs that downstream tasks must parse, (3) actionable failure feedback—when code executes but produces wrong/incomplete results, what specifically failed?
> > >
> > > **Our approaches are complementary:** You could execute actions as Python code while defining expected typed entities for validation. See Section 2, paragraph 2.
> > >
> > > ### On Reasoner vs Entity Extractor
> > >
> > > **Entity Extractor:** Tool call tasks only
> > > - Input: Unstructured tool response
> > > - Purpose: Parse verbose external outputs
> > >
> > > **Reasoning Agent:** Cognitive tasks
> > > - Input: Resolved task parameters
> > > - Purpose: LLM analysis producing reasoning + entities in single call
> > >
> > > Cannot combine due to: (1) different inputs (tool responses vs parameters), (2) different timing (post-tool vs during-execution), (3) different purpose (parsing vs producing). See Section 3.3.1 and Appendix A.3-A.4.
> > >
> > > ### On Computational Efficiency
> > >
> > > We apologize for the confusion in our original metrics (for both tokens and LLM calls). The corrected metrics (Section 5.3, Figure 3) shows:
> > > - Higher LLM call frequency (extraction + replanning overhead)
> > > - Superior token efficiency per call (0.8-1.2K vs ReAct's 2.3-2.7K)
> > > - Explicit tradeoff: more calls for higher success rates
> > >
> > > We believe these revisions substantially strengthen the contribution. The empirical results demonstrate that entity-based plan structure provides measurable reliability improvements over existing approaches including CodeAct.
> > >
> > > We would be grateful if you would reconsider your assessment.
> > >
> > > Sincerely,
> > > The Authors

---

### Note · Program_Chairs · 2026-01-17
**Submission Desk Rejected by Program Chairs**

The following references in this submission do not refer to real documents and/or have major errors in bibliographic information:

 Kiran Lee, Tianyu Zhang, Minjoon Seo, Hannaneh Hajishirzi, and Noah A. Smith. Loft: Long-context frontiers for large language models. arXiv preprint arXiv:2406.12832, 2024.